# Ground-penetrating radar imaging reveals glacier's drainage network in 3D

Gregory Church[1,2], Andreas Bauder[1], Melchior Grab[1,2], and Hansruedi Maurer[2]

[1]Laboratory of Hydraulics, Hydrology and Glaciology (VAW), ETH Zurich, Zurich, Switzerland
[2]Institute of Geophysics, ETH Zurich, Zurich, Switzerland

**Correspondence:** Gregory Church (church@vaw.baug.ethz.ch)

**Abstract.**

Hydrological systems of glaciers have a direct impact on the glacier dynamics. Since the 1950's, geophysical studies have provided insights into these hydrological systems. Unfortunately, such studies were predominantly conducted using 2D acquisitions along a few profiles, thus failing to provide spatially unaliased 3D images of englacial and subglacial water pathways. The latter has likely resulted in flawed constraints for the hydrological modelling of glacier drainage networks. Here, we present 3D ground-penetrating radar (GPR) results that provide high-resolution 3D images of an alpine glacier's drainage network. Our results confirms a long-standing englacial hydrology theory stating that englacial conduits flow around glacial overdeepenings rather than directly over the overdeepening. Furthermore, these results also show exciting new opportunities for high-resolution 3D GPR studies of glaciers.

## 1 Introduction

Glacier movement is the combination of internal ice deformation and basal motion. Basal motion comprises both ice sliding over the glacier bed and the deformation of subglacial till (Cuffey and Paterson, 2010). Sliding at the ice-bed interface is responsible for high ice flow velocities (often 100%-400% faster than the annual mean flow velocity (Macgregor et al., 2005; Bingham et al., 2006; Bartholomew et al., 2010; Tuckett et al., 2019)) as a result of reduced friction at the ice-bed interface (Bartholomaus et al., 2008). This reduction of friction is caused by the subglacial drainage network that lubricates this interface and increases subglacial water pressure, thereby either weakening subglacial sediments (Schoof, 2010) or lubricating the hard bedrock. In alpine glaciers and in Greenland, the subglacial drainage network is fed from surface meltwater that is routed through the englacial drainage network (Fountain and Walder, 1998). At the beginning of the melt season and with an increased availability of surface melt water, the subglacial drainage network often experiences an increase in water pressure, since the drainage network cannot adapt quickly enough to the increase in meltwater influx (Iken et al., 1983). During these periods with increased subglacial water pressure, changes in the glacier's sliding velocity are often observed (Gudmundsson et al., 2000; Sugiyama and Gudmundsson, 2004; Macgregor et al., 2005), and it has been widely documented that increased glacier sliding velocities have the potential to increase the glacier's mass loss (Zwally et al., 2002; Joughin et al., 2008; Schoof, 2010). Whilst the existence of these variations in ice flow velocities are undisputed, there is limited observations of the hydrological system's

geometry and its temporal variations, thus hampering a deeper understanding of these seasonal variations (Hart et al., 2015; Church et al., 2020).

The glacier's hydrological system can be probed using a variety of methods. Direct observations have been made from boreholes (Fountain et al., 2005), tracer testing (Nienow et al., 1996), speleology (Gulley, 2009; Gulley et al., 2009; Temminghoff et al., 2019) and geophysical measurements. The latter include predominantly active (Peters et al., 2008; Zechmann et al., 2018;
Church et al., 2019) and passive (Podolskiy and Walter, 2016; Lindner et al., 2019; Nanni et al., 2020) seismic measurements or ground-penetrating radar (GPR) measurements (Moorman and Michel, 2000; Stuart et al., 2003; Irvine-Fynn et al., 2006; Harper et al., 2010; Bælum and Benn, 2011; Hansen et al., 2020). Most glaciological GPR studies, published so far, relied on two-dimensional (2D) data, where GPR measurements were acquired along profiles. 2D data sets are typically unable to image complex subsurface structures, such as glacier drainage networks, and the resulting interpretations may thus be inconclusive.
For small-scale targets, such as archaeology sites (Böniger and Tronicke, 2010) and shallow fault zones (McClymont et al., 2008), 3D GPR surveys have established themselves as a powerful and efficient tool to image complex subsurface structures. Moreover, 3D radar surveys have also been leveraged on large scale applications to investigate extraterrestrial ice bodies (Putzig et al., 2018). 3D GPR surveys are composed of densely spaced multiple line-by-line 2D GPR profiles that are collectively processed and are able to avoid both sampling bias in the profile's in-line direction and aliasing in the cross-line direction. Spatially
unaliased 3D GPR data sets (i.e., datasets with a data point spacing smaller than the dominant wavelength of the GPR signals (Sheriff and Geldart, 1995; Grasmueck et al., 2005)) are rare in glaciological applications. This is unfortunate, because 3D GPR provides subsurface images that can be viewed from arbitrary directions, thus allowing for an unequivocal interpretation. Furthermore, 3D GPR glaciological surveys can provide high spatial resolution imaging of the glacier's drainage network. Such an approach would be particularly useful for glacier drainage networks, and should be feasible because of the strong
reflections caused by the very pronounced electrical impedance contrasts at ice/water interfaces (Reynolds, 1997).

To date, there are only a small number of glaciological studies leveraging 3D GPR to gain insights into the glacier's hydraulic system. 3D GPR data were used by Harper et al. (2010) to investigate basal crevasses and the subglacial hydraulic network on Bench Glacier, Alaska. More recently, Egli et al. (2021) acquired and processed 3D GPR surveys over two Swiss glaciers and successfully imaged the subglacial drainage network with the analysis of the reflected GPR amplitudes. Several glaciological
studies (Schaap et al., 2019; Hansen et al., 2020; Church et al., 2020) have used multiple 2D GPR profiles to investigate the englacial drainage structure in both cold and temperate ice; however, as far as we are aware, there are no studies leveraging 3D GPR studying in order to image an englacial drainage network.

The direction that meltwater flows under and within a glacier is driven by the spatial gradient of the hydraulic potential, outlined by Shreve (1972), where the hydraulic gradient is a function of both the water pressure gradient and the elevation
potential gradient. Subglacial water flows along the hydraulic gradient and upon meeting an overdeepening, Lliboutry (1983) hypothesised the water flows along the glacier's flank as so-called gradient conduits, therefore avoiding the deepest part of the overdeepening (Cook and Swift, 2012). According to Lliboutry's theory, these conduits should be located at the same altitude as the lowest point of the riegel that produces the overdeepening. This hypothesis has a direct consequence on glacier sliding

theory as no subglacial waterways should exist in the overdeepening and thereby potentially altering the glacier's sliding velocity.

In this study there are three main objectives, namely to

- demonstrate the feasibility of and opportunities offered by 3D GPR imaging on glaciers.

- provide much needed hydrological observations to determine whether they are in agreement with long-standing glacier hydraulic theory regarding englacial water pathways around overdeepenings as described by Lliboutry (1983).

- provide an insight into future opportunities for high-resolution radar studies of glaciers.

## 2 Survey Site & Previous Work

The 3D GPR acquisition was conducted on Rhonegletscher, a temperate glacier located in the central Swiss Alps (Fig. 1a). The Rhonegletscher is representative for the majority of European mountain glaciers with regard to its temperature distribution, ice dynamics and size (GLAMOS, 2017; Beniston et al., 2018). Rhonegletscher is the sixth largest glacier in the Swiss Alps (length: 8 km, surface area: 15.5 km$^2$ as of 2015 (GLAMOS, 2017)) and is heavily exposed to glacier melt due to the changing climate. Over the last decades, the glacier has continued to thin and it is currently retreating. As a result, a proglacial lake fed by the drainage network has been forming at its terminus since 2005 (Tsutaki et al., 2013).

The 3D GPR survey was motivated by the results of earlier 2D surveys. In 2017, a strong englacial reflection was identified from both active 2D seismic data and 2D GPR measurements and reflection analysis resulted in a water-filled englacial conduit being identified (Church et al., 2019). In the 2018 melt season a borehole was drilled into the englacial conduit and a borehole camera observed a water-filled and actively flowing hydraulic network. During 2018 and 2019 repeated GPR measurements on a coarse grid of 2D lines (Church et al., 2020) provided initial imaging of a potential drainage network and its seasonal changes. The repeated measurements provided evidence that the englacial conduit was 0.4 m $\pm$ 0.35 m thick, 17.5 m $\pm$ 8.5 m wide and highlighted seasonal variations of an actively flowing englacial conduit. However, the surveys failed to image the larger extent of the drainage network and determine whether it connects to a subglacial hydraulic network. Additionally, it had limited spatial resolution due to the 2D nature of the GPR data. The englacial network is located within an overdeepening and therefore provides a suitable candidate to determine whether the network is in agreement with current hydraulic theory.

## 3 Methods

### 3.1 Data Acquisition

To detect and characterise the drainage network located at the glacier's tongue, we acquired a high-resolution 25 MHz 3D GPR survey. Between 15[th] July 2020 and 23[rd] July 2020, GPR data were acquired over the area expected to harbour the englacial conduit network (Fig. 1b).

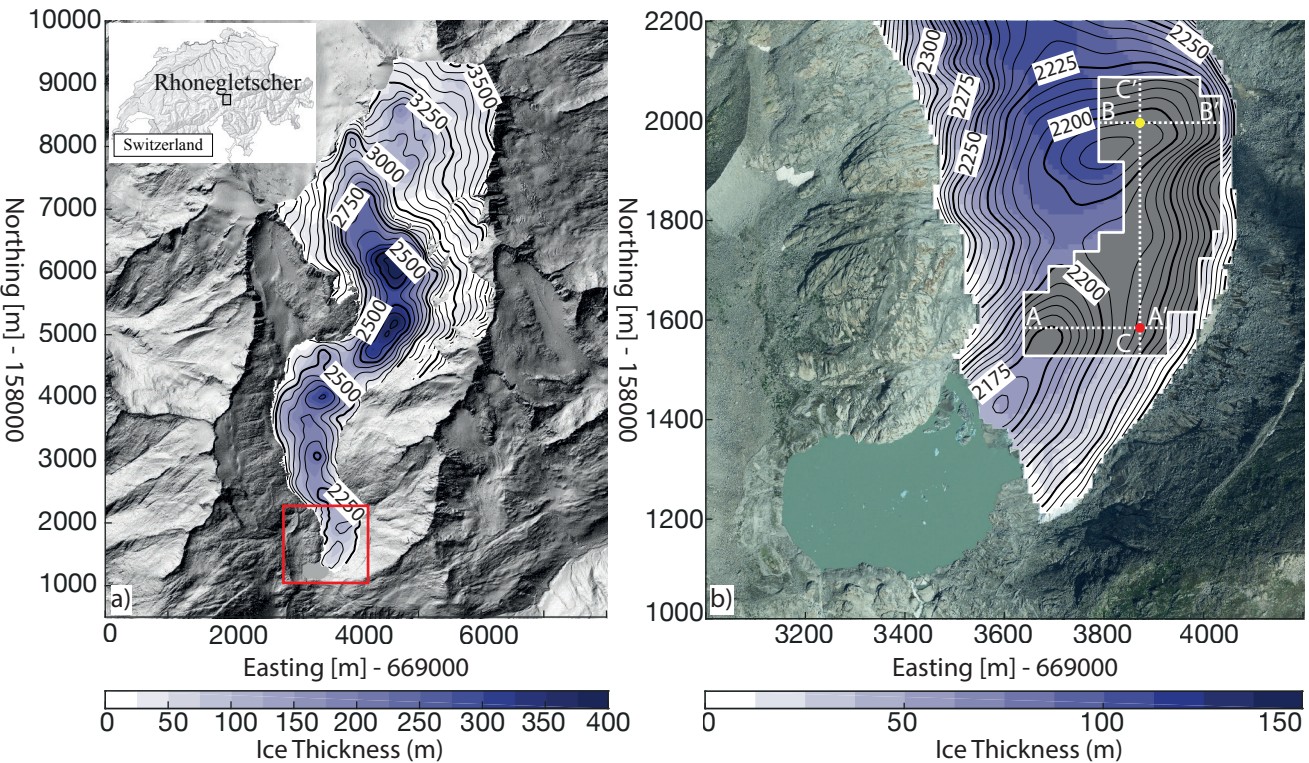

**Figure 1.** a) Rhonegletscher ice thickness in 2019 (colours) and bedrock elevation (contours) estimated using the GlaTe model from 59 interpolated radar profiles acquired between 2003 and 2008 as described by Grab et al. (2021). The red box represents the zoomed area for panel (b). b) Lower ablation zone of Rhonegletscher showing ice thickness (colour) and bedrock elevation (contours). The grey polygon represents the 3D GPR survey site, three GPR profiles A-A', B-B' and C-C' are shown in Figure 2 and the crossing points of the GPR profiles are represented by a yellow and red dot. Co-ordinates for all plots are local swiss co-ordinates LV03. Orthophoto was provided by Swiss Federal Office of Topology: Reproduced by permission of swisstopo (JA100120), ©2020 swisstopo (JD100042)

The survey covered an area of 140,000 m$^2$, within which the ice thickness varied between 25 and 110 m and where the glacier bed forms a distinct overdeepening (Fig. 1b). The common-offset GPR data were collected using a Sensor & Software PulseEkko® system with an antenna separation of 4 m and carried by hand at approximately 1 m above the glacier ice surface. The sampling rate of the GPR system was 1 GHz, giving a time resolution of 1 ns. The use of a large sampling frequency allows small topographical changes from trace to trace of <0.2 m to be observed (King, 2020). A GPR stacking of 4 improved the signal-to-noise ratio and allowed the GPR data to be acquired with average walking speed of approximately 0.4 m per GPR trace. For all GPR lines, a high precision global navigation satellite system (GNSS) continuously recorded the x, y, z coordinates of the centre point between the transmitting and receiving antennas every second. The average accuracy of the GNSS during GPR acquisition was 0.008 m.

The survey area was covered with 281 2D GPR profiles, resulting in a total profile length of approximately 85 km. The GPR data for our 3D processing workflow were collected perpendicular to the ice flow direction with 2 m interspacing between 2D GPR profiles. The line spacing was chosen, such that the diffractions and reflections within the ice body would not become

aliased for our antenna frequency of 25 MHz. To ensure data were consistent across the duration of the survey, two GPR lines were always repeated from the previous day's acquisition. Furthermore, six orthogonal profiles were collected along the glacier flow for quality control purposes.

## 3.2 Data Processing

All GPR common offset data were processed using a combination of an in-house MATLAB-based toolbox (GPRglaz) (Grab

et al., 2018) and SeisSpace ProMAX 3-D. The processing was based upon a typical 3D seismic data processing workflow. Initially, the GPR data were assigned to their corresponding GNSS data. Since the GNSS data were recorded every second and GPR data were recorded approximately every 0.3 seconds, the GNSS data were linearly interpolated to provide the same temporal resolution as the GPR data. The data were then corrected for time zero, and a Butterworth bandpass filter was applied in order to suppress any noise outside the GPR frequency band and thus to increase the signal-to-noise ratio. Overlapping

GPR data between different acquisition days were used to investigate whether a data matching filter was required. However, no amplitude matching was required, due to the fact that the GPR equipment used produced stable and repeatable GPR data. Subsequently, the data were 3D binned using a master grid of 2 m between GPR profiles (inline spacing) and 0.5 m between GPR data points within the profile (crossline spacing). 3D binning consists of assigning each GPR trace to the closest bin centre. As a result of the variable walking speed and walking around crevasses during acquisition, a number of bins had more

than a single GPR trace assigned (known as over-fold), whereas other bins (i.e., bin located within a crevasse) were empty. GPR data points were removed from bins that had more than one GPR data point and therefore, this resulted in the bins containing either a single GPR data point or no GPR data point. Such a processing step is required in order not to leave an amplitude imprint on the data during the GPR interpolation stage. The resulting single fold GPR data were interpolated such that all bins were filled and the GPR data were positioned at the centre of their bins.

A 3D Kirchhoff migration algorithm re-positioned the reflected and diffracted signals to their correct subsurface location. The Kirchhoff migration algorithm was performed using an EM wave propagation velocity of ice (0.167 m ns$^{-1}$) as velocities between 0.165 and 0.170 m ns$^{-1}$ were confirmed for this site by Church et al. (2020). Furthermore, the migration algorithm corrected for amplitude losses from geometrical spreading, whereas no correction for radiation pattern was applied. Prior to interpretation, a topographyic correction, an amplitude correction using a Q attenuation compensation (Irving and Knight,

2003), and a second Butterworth bandpass filter was applied to further improve the signal-to-noise ratio and to suppress the high frequency noise artificially increased by the Q-compensation correction. Finally, the data were converted from the time to depth domain using a constant velocity of 0.167 m ns$^{-1}$.

The 3D interpretation was performed in dGB Earth Sciences OpendTect. The ice-bed interface was manually picked, linearly interpolated, smoothed and constrained to within the survey area. Secondly, the drainage network was located from the 3D

processed GPR data as the strongest continuous coherent reflection and manually picked with aid of previous GPR, seismic

and borehole studies (Church et al., 2019, 2020). Subsequently, the drainage network was linearly interpolated and smoothed. In order to ensure the picked drainage network encompassed the entire observable drainage network in the GPR survey area, GPR elevation slices were investigated to locate strong englacial and subglacial reflections that could represent a water-filled drainage network (Fig. 3) and the root-mean squared amplitude was extracted between the surface and the ice-bed interface.

Finally, the reflected GPR amplitudes were extracted from both the ice-bed interface and the drainage network by using a 2 m window (±1 m) centred around the feature. During interpretation care was taken along the edges of the survey as a result of GPR data migration edge effects.

## 4  Results

### 4.1  Overview of GPR results

Displaying 3D models adequately is generally a non-trivial task. Below, we discuss the GPR results using a variety of vertical and horizontal cross sections. In our view, such data sets are represented best in form of movies showing scans along different directions. We therefore highly encourage the readers to check the digital supplement.

Selected 2D profiles of the 3D GPR data cube are shown in Figure 2. A water-filled englacial conduit can be identified as a continuous specular strong reflector (Fig. 2). The majority of the ice-bed interface is identifiable as a weak reflection (Fig 2),

indicating that subglacial water is not present. However, in isolated areas, the ice-bed interface has been identified as strong ice-bed reflections (Fig. 2c) and thereby indicating the presence of subglacial water.

The lateral extent of the englacial and subglacial network can be characterised by analysing horizontal slices of the 3D GPR data cube at different elevations above mean sea level. A slice at 2216 m.a.s.l shows a strong meandering reflection in the northern part of the survey (Fig 3a). The strong reflection is traceable on the eastern edge of the survey before fading

out towards the southern edge. At 2213 m.a.s.l (Fig 3b), there is a continuation of the strong reflection, but it becomes more diffusive in the central part of the survey area. At 2210 m.a.s.l (Fig. 3c), we observe another strong meandering reflection that heads southwards towards the terminus of the glacier. There is an approximately 6 m topographic difference between the drainage network in the survey's northern edge and the southern edge indicating that the imaged drainage network has a shallow global inclination along the flow (<1 degree).

### 4.2  Spatial distribution of drainage network


The 3D GPR survey imaged an active meltwater drainage network within the survey boundary. It comprises both, an englacial and subglacial drainage network. The entire detectable drainage network was identified from the final processed GPR data, and reflected amplitudes from the drainage network were extracted (Fig 4a) as detailed in the data processing section. The conduit network can be delineated by areas of high amplitude (yellow in Fig. 4a). Furthermore, the drainage network can be split into

four separate components labelled in Fig. 4b:

(A)  A meandering well-defined englacial conduit spanning the overdeepening oriented perpendicular to the glacier flow,

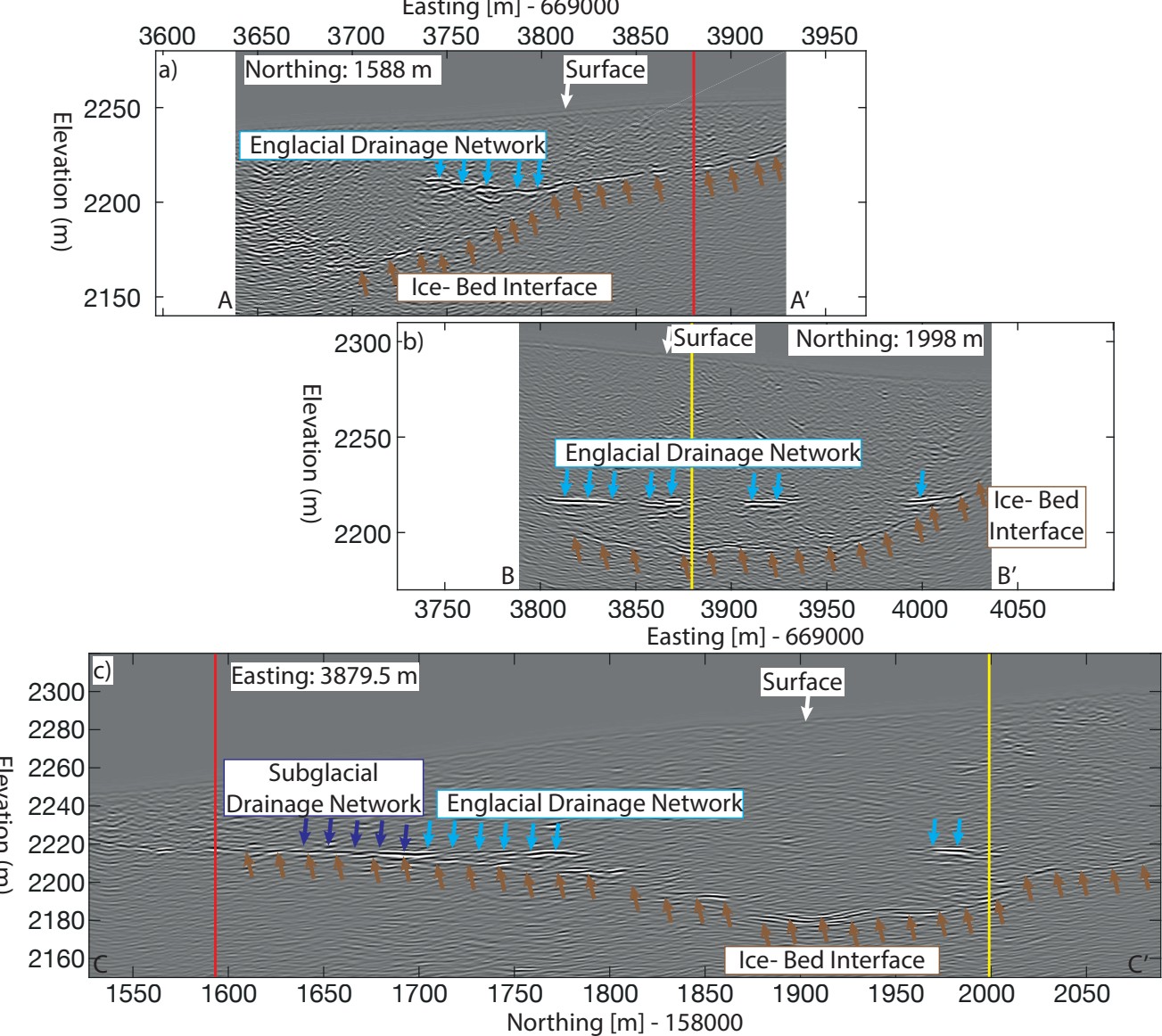

**Figure 2.** a) GPR inline profile (perpendicular to ice-flow direction), the glacier surface, drainage network and basal interface are marked and the red line represents the crossing point for profile c). b) GPR inline profile (perpendicular to ice-flow direction), the yellow line represents the crossing point for profile c). c) GPR crossline profile (parallel to ice-flow direction). The locations of the profiles are shown in Figure 1b and 3

(B)  An englacial conduit oriented parallel to the glacier flow traversing the overdeepening and flows alongside the glacier's flank,

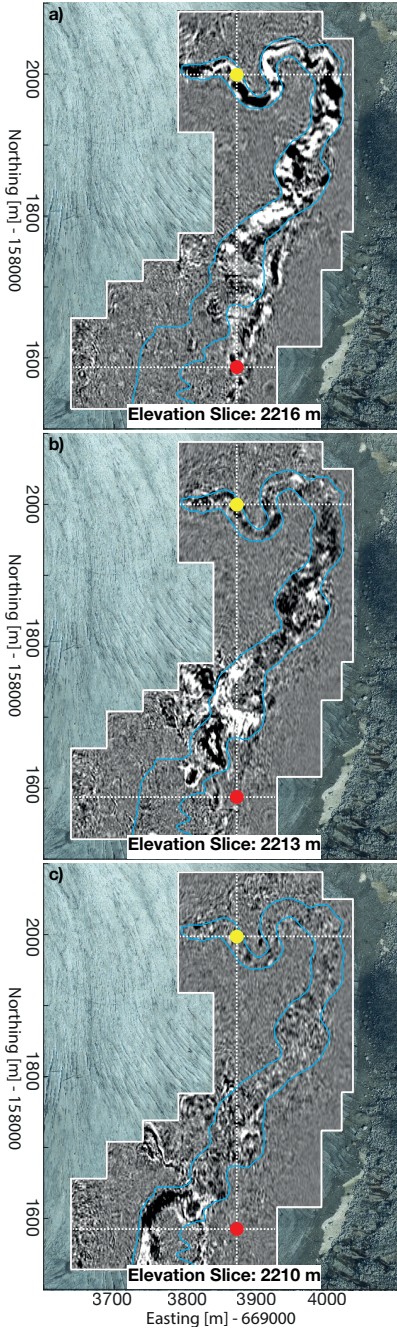

**Figure 3.** a) Depth slice through the GPR 3D data cube at 2216 m amsl. b) 2213 m amsl. c) 2210 m amsl. The blue line represents the outline of the drainage network. The white dashed line represents the GPR profiles from Figure 2 and the red and yellow dot represents their crossing points.

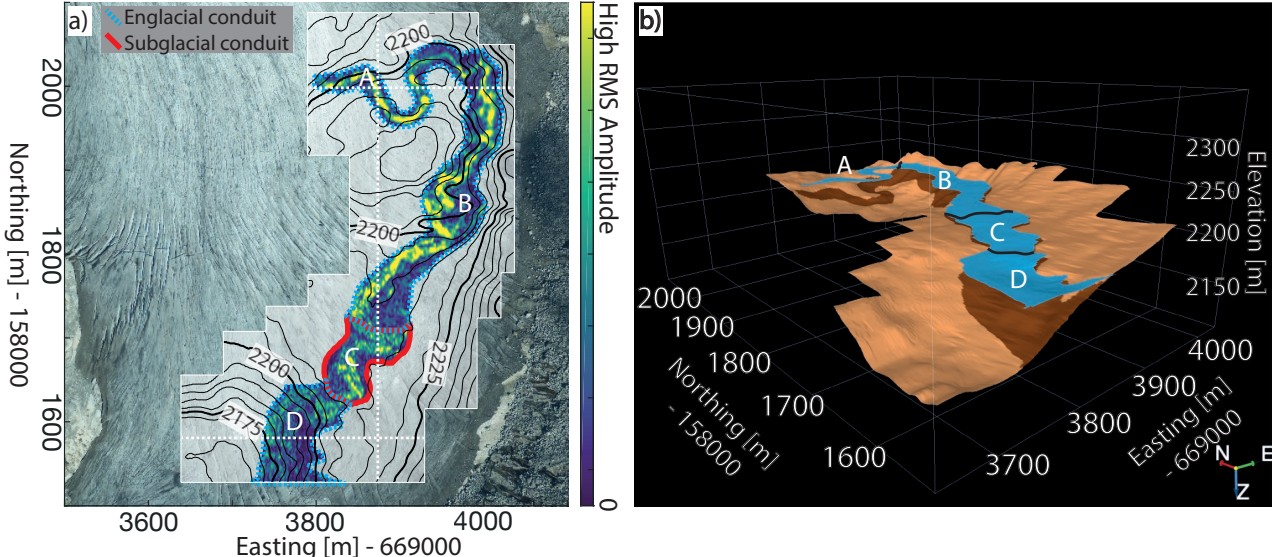

**Figure 4.** a) The root-mean-squared amplitude, extracted from the picked glacial drainage network, within ±1 m of the drainage network. Contours represent the basal topography picked from the 3D GPR processed data. Colours of the drainage network represent the reflected amplitudes and areas of high amplitude indicate the presence of water. b) 3D view of basal interface (brown) and drainage network (blue). The drainage network is split into 4 components labelled A to D and referred to in the text.

(C) The englacial conduit in B connects to the subglacial drainage system, the subglacial drainage network consists of a single main conduit (Fig 4a), that has a sinusoidal nature,

(D) The subglacial conduit in C encounters a basin and encounters a diffusive network of englacial conduits towards the terminus of the glacier that are poorly imaged.

Given the glacier ice flow direction (N-S) and the ice-thickness distribution, the water in the conduit is expected to flow from north to south. The high-resolution results allow the width of the drainage network to be examined and the uncertainty is attributed to the post-migration lateral resolution. The mean width of the northern sinusoidal englacial conduit, which flows across the overdeepening (Fig. 4b Section A), is $8 \pm 1.7$ m. As the network flows southwards around the overdeepening (Fig. 4b Section B) the width increases to $11 \pm 1.7$ m. Furthermore, the mean width of the subglacial drainage conduit (Fig. 4b Section C) is $17 \pm 1.7$ m, and finally at the southern edge of the survey site, the mean width of the diffusive englacial drainage network is $25 \pm 1.7$ m (Fig. 4b Section D). The thickness of the conduit in section A has previously been investigated in Church et al. (2020), and it was found to be at the limit of the 25 MHz GPR vertical resolution at 0.4 m, when assuming the conduit is water-filled. The conduit thickness in sections B, C and D are also at the limit of the vertical resolution as only a single reflection is visible (Fig. 2c). If the conduit thickness was beyond the vertical resolution, two separate englacial reflections would be visible representing the top and bottom of the conduit. Consequently, the channels throughout the study area are thinner than 0.4 m, and therefore, their shape is significantly smaller in the vertical direction than lateral.

 ## 4.3 Basal reflected amplitude

The amplitude of the ice-bed interface was also extracted. This provided insights into the basal conditions such as bedrock type and whether subglacial water is present. In the southern region of the survey site, the ice-bed interface reflection has an identical amplitude and spatial distribution to the drainage network (Fig. 5a and c white arrows), thereby indicating this area is identical to the drainage network identified in Figure 4a. In the northern region of the survey site, there are numerous isolated high

amplitude basal reflections (Fig. 5a), mostly situated within localised flat areas (Fig. 5 b red arrows) and most likely indicating a pooling of water. In addition, ubiquitous areas of high amplitude basal reflections are present along the southern region of the survey, that could indicate the presence of subglacial water (Fig. 5a and c pink arrows). In comparison to the isolated patches in the northern region, these high amplitude basal patches in the southern region appear to be partially connected to each other indicating the possibility of an additional subglacial drainage system away from the main drainage network. These areas were

not picked as part of the main drainage network due to their different data characteristics (not being connected to the main drainage network - white arrows in Fig. 5).

## 4.4 Comparison of 2D and 3D GPR processing

2D GPR surveys along single profiles are the current approach in glaciological applications, although such data sets have imaging limitations. Besides being unable to provide high resolution 3D subsurface images, 2D GPR techniques assume

all reflections originate from the vertical plane of the acquisition. Complex englacial structure or basal geometry can result in a reflection originating from outside of the acquisition plane, in turn resulting in distortions to the final processed GPR image. Figure 6 shows an example ray path causing such a distortion as a result of off-nadir reflections. These distortions are particularly severe for complex geometry of alpine glaciers and even with the use of densely space 2D GPR lines that are processed independently with a 2D migration, these features will not be correctly positioned. Recent studies have experimented

with acquiring swaths of radar profiles using fixed-wing aircraft to bridge the gap between 2D and 3D acquisition geometries by accounting for these off-nadir reflections (Holschuh et al., 2020).

3D GPR acquisition and processing are able to fully mitigate these distortions. The Rhonegletscher GPR data cube is the product of a 3D processing workflow and with the employment of 3D migration over conventional 2D migration, the off-nadir distortions are removed and an improvement in lateral resolution is gained. A 3D migration effectively collapses the

Fresnel zone in both inline and crossline directions, thereby reducing the lateral resolution to the wavelength of the EM-wave propagating through ice. This lateral resolution leads to improvements in subsurface imaging, as two closely laterally-separated reflectors are able to be imaged as two individual reflectors.

A comparison between GPR data processed with two different workflows (2D GPR workflow (Figure 7a) and 3D GPR workflow (Fig. 7b)) highlights the imaging differences on both the englacial conduit network and the ice-bed interface. Gen-

erally, both workflows produce similar subsurface images however, there are subtle differences that indicate a less ambiguous interpretation with the 3D GPR workflow. The ice-bed interface in the 3D GPR data cube has increased reflector continuity in comparison to the 2D workflow (Fig. 7 brown arrows). Furthermore, the englacial conduit imaged using a 3D GPR work-

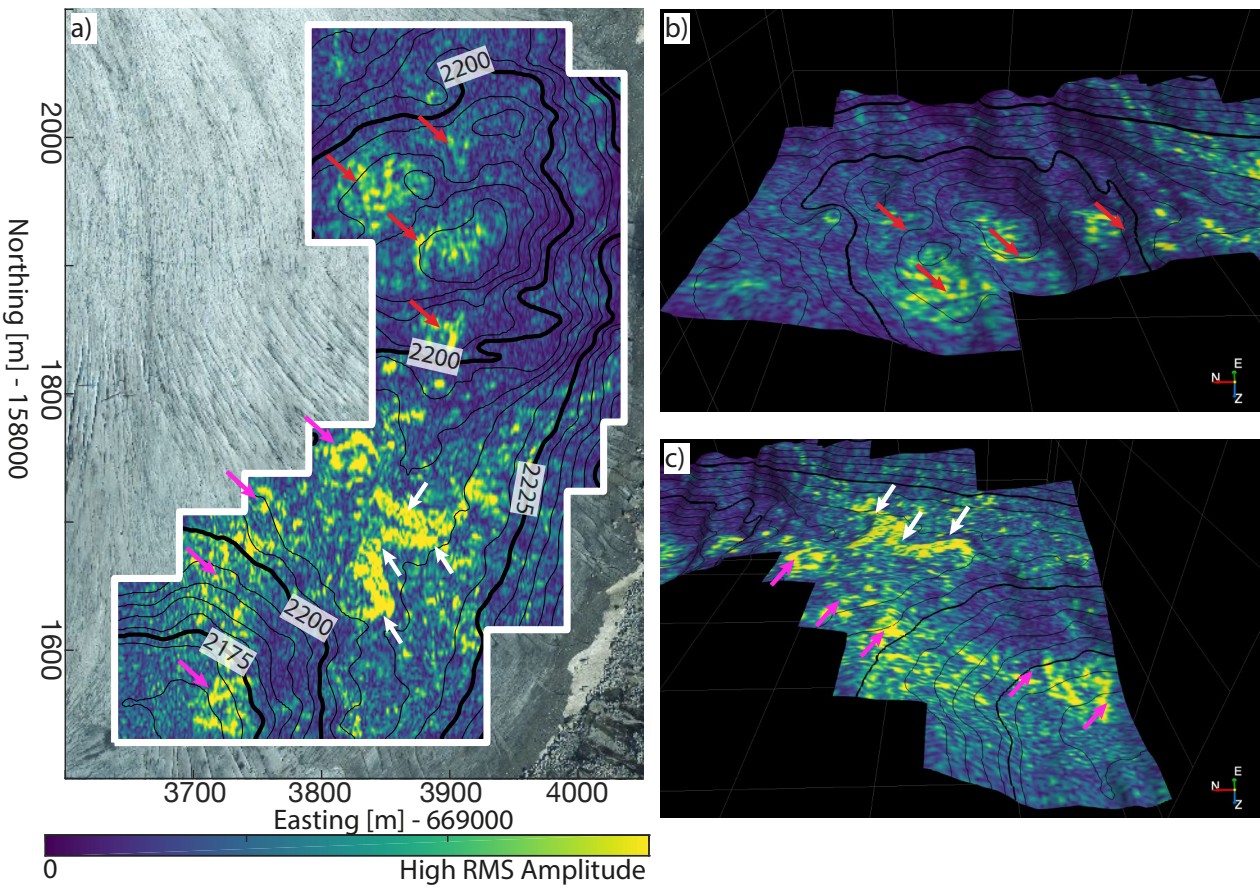

**Figure 5.** a) Map view of the extracted basal root-mean squared amplitude within a $\pm 1$ m window. Contours represent the basal topography picked from the 3D GPR processed data. b) Extracted basal amplitude in the northern part of the survey highlighting isolated high amplitude basal reflections situated in localised flat zones, c) extracted basal amplitude in the southern part of the survey highlighting connected high amplitude basal reflections. The red arrows represent isolated water cavities along basal interface, the white arrows represent the main drainage network detected in Fig. 4. The pink arrows indicate the presence of subglacial water flow away from the main drainage network identified in Fig 4.

flow has fewer artefacts and is absent of events that are incorrectly intersecting the englacial conduits (Fig. 7 blue arrows). Additional 2D and 3D GPR comparisons are provided as supplement figures (Fig. S1).

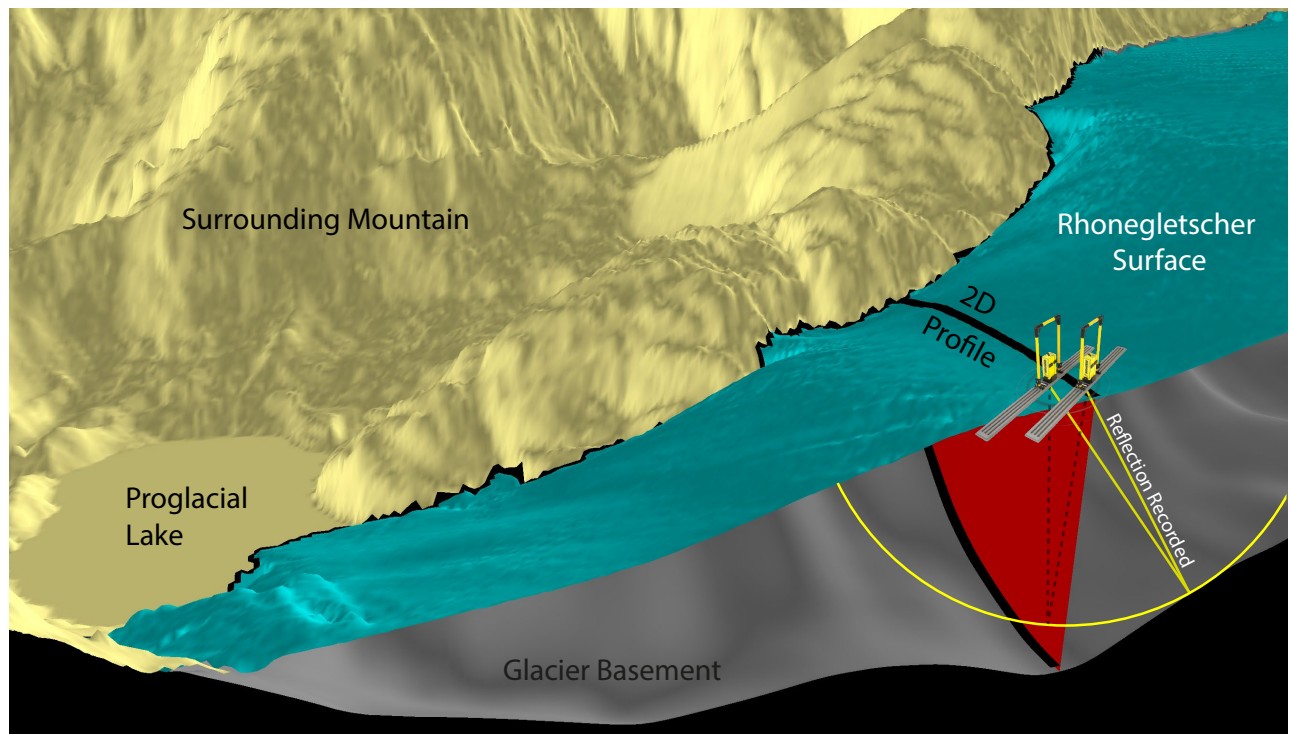

**Figure 6.** Lower portion of Rhonegletscher showing an example GPR data point acquired. The GPR antennas are located on the surface and the yellow lines represent the shortest ray path for a reflection from the glacier basement. The dashed black ray path represents the basement imaging point when performing 2D GPR processing. This ray path does not correspond to the true basement position but the out-of-plane basement reflection point. This type of ray path is known as an off-nadir reflection.

## 5 Discussion

### 5.1 Geometry of drainage network

Alongside Harper et al. (2010) and Egli et al. (2021), this study is one of a few times that a glacier's drainage network is imaged in 3D with GPR data, thus providing high-resolution information of the geometry from such a system. The Rhonegletscher's drainage network identified in this study has a meandering nature throughout the survey area, with an increasing network width towards the terminus of the glacier. Moreover, it consists of a single dominant conduit that alternately flows through englacial and subglacial channels, known as Röthlisberger channels (Röthlisberger, 1972). Such a drainage network is known as an efficient channelised network. Studies from both polar (Chandler et al., 2013) and temperate (Nienow et al., 1996, 1998; Mair et al., 2002) glaciers have shown that early in the melt season, the glacier's drainage network is slow and inefficient. Typically, it evolves into a fast channelised drainage network just before the peak of the glacier's discharge. Since the peak discharge for

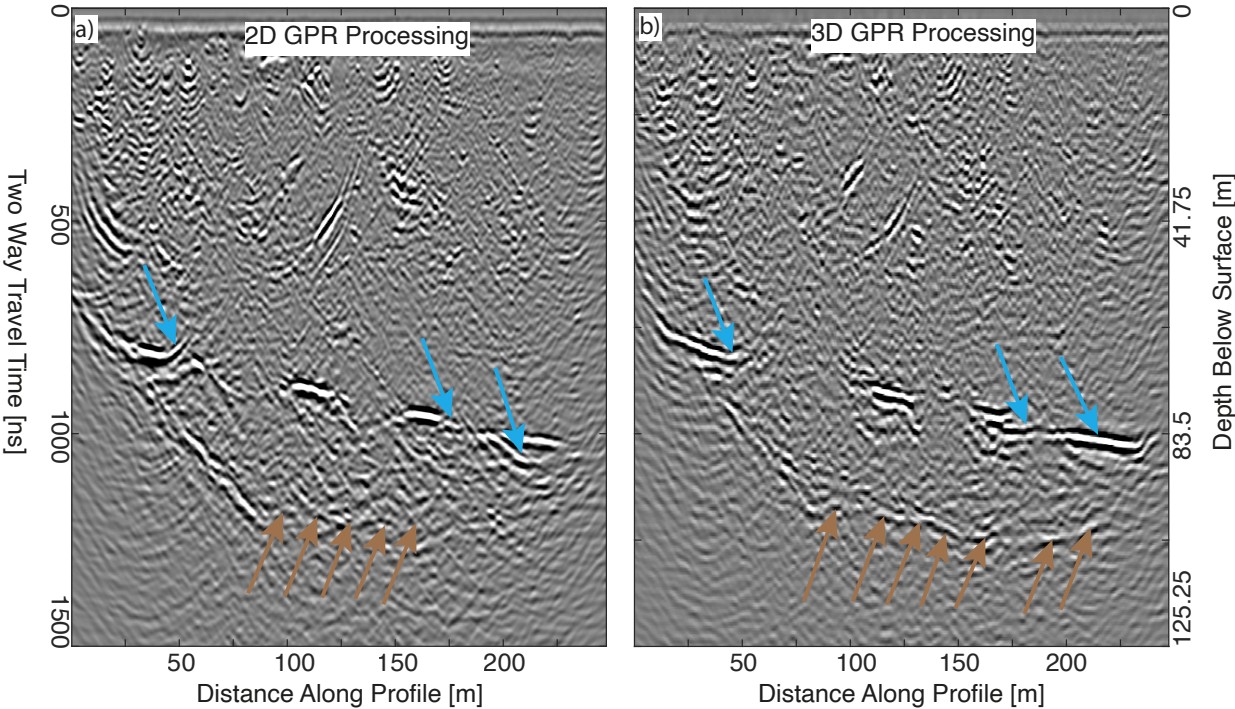

**Figure 7.** a) A single GPR data profile processed using with a 2D GPR workflow. b) Single line extracted from multiple GPR profiles processed using a 3D GPR workflow.

Rhonegletscher is typically mid-August (GLAMOS, 2017) (one month after data acquisition), the drainage network is expected to be in a channelised configuration.

The theoretical shape of englacial drainage conduits is circular, however the drainage network imaged on Rhonegletscher is up to 60 times wider than its thickness when water-filled (according to both borehole observations and reflected GPR polarity). Such an observation contradicts the theory of circular conduit cross-sectional shape proposed independently by

Shreve and Röthlisberger (Röthlisberger, 1972; Shreve, 1972) but is in line with the further development by Hooke et al. (1990). The latter author concluded that conduit's are broad and low based upon measured and calculated subglacial water pressures on Storglaciären, Sweden. Such channels can directly impact the glacier dynamics as Hooke-channels can lead to increased hydraulic friction and thus higher water pressure than theorised R-channels. This increase in hydraulic friction is not only a result of the channel's shape but also due to higher closure rates of the conduit. Thereby, the impact on ice dynamics is that

such a configuration would support higher sliding velocities. Furthermore, Werder et al. (2010) found that the hydraulic friction interpreted from tracer experiments could be well explained by assuming low and broad channels (i.e., Hooke channels).

Lliboutry (1983) hypothesised that when water encounters an overdeepening, the water flows along the glacier's flank as so-called gradient conduits, therefore avoiding the deepest part of the overdeepening (Cook and Swift, 2012). According to Lliboutry's theory, these conduits should be located at the same altitude as the lowest point of the riegel that produces the

overdeepening. The 3D GPR data suggest that in the case of Rhonegletscher, the flow paths indeed route meltwater around the overdeepening rather than across it (Fig. 4b Labelled: B). Similarly, the elevation of the englacial conduit that is routing meltwater around the overdeepening coincides with the elevation of the riegel and also the proglacial lake level. These observations support the long-standing theory by Lliboutry (1983), but was never verified by field observations.

Subglacial and englacial water flows as a response of changing hydraulic potentials. This hydraulic potential can be estimated
by assuming spatially uniform flotation fraction as described in Flowers and Clarke (1999). The imaged Rhonegletscher's drainage network followed the gradient of the hydraulic potential and not along a single hydraulic potential contour.

## 5.2 Water accumulation in temperate glaciers

In addition to the detection of the primary drainage network, the 3D GPR data provided possible evidence of subglacial water accumulation. GPR-based detection of large amounts of subglacial water, such as subglacial lakes is well established (Ridley
et al., 1993; Siegert et al., 2005; Palmer et al., 2013), but their spatial extent is often unclear as a result of the limitations of 2D GPR surveys.

From our 3D GPR data set, we are able to delineate high-resolution lateral changes to the basal interface. Furthermore, 3D GPR has the potential to identify smaller subglacial water accumulations, such as expected to occur within water-filled cavities. Subglacial cavities can form, when the sliding ice uncouples from the glacier bed as a result of either rapid glacier sliding or
pronounced bed roughness (Nye, 1970). Two types of subglacial cavity system are generally distinguished – isolated cavities and linked cavities – and both cavity systems alter the glacier's dynamics (Lliboutry, 1976, 1979; Hoffman et al., 2016; Rada and Schoof, 2018). The high amplitude reflections along the basal interface (Fig 5a) likely represent either water accumulations along basal bedrock or saturated sediments both of which appear to be isolated from each other. However, saturated sediments are unlikely on Rhonegletscher as a result of outcrops showing a granite bedrock with little sediment cover and borehole
observations within the survey area showing a hard bedrock basal interface (Church et al., 2019). Furthermore, the location of these high amplitude basal reflections can be explained from the hydraulic potential when assuming low subglacial water pressure (Fig. S2a). On the other hand, when assuming high subglacial water pressure these high amplitudes are located along the hydraulic gradient (Fig. S2d). Due to the nature of the diurnal subglacial water pressure on Rhonegletscher (Sugiyama et al., 2008) it is therefore likely that these high amplitude basal reflections are indicative of potentially isolated water-filled
cavities forming an inefficient drainage network.

High amplitude basal reflections could also result from air filled cavities. If an air-filled cavity existed, the recorded EM reflection would have opposite polarity to a reflection caused by a water-filled cavity or hard bed. In the case of the Rhonegletscher 3D GPR data set the phase of the basal reflection remained consistent across the survey suggesting that there are no imaged air-filled cavities within the survey area. Furthermore, it is interesting to note both an inefficient drainage network and
an efficient network can coexist in overdeepenings (Hooke et al., 1990; Rada and Schoof, 2018). Although our data provide an instantaneous image of these systems, repeated 3D GPR surveys could also yield insights into their temporal dynamics.

## 5.3 Future of 2D and 3D GPR within glaciology

A 3D approach as presented within this contribution is feasible and highly beneficial over 2D GPR for detecting and quantifying dimension of a glacier's hydrological network. For large-scale investigations in Greenland and Antarctica, it will be more challenging to conduct 3D GPR surveys as a result of their spatial distribution and therefore, 2D GPR acquisition will likely continue to prevail. However, future radar surveys could be complimented with the use of 3D GPR acquisition, in order to reduce the ambiguity of interpretations in places of interest. The 2D and 3D GPR data processing comparison (Fig. 7 and Fig. S1) highlights the subtle difference in the advantage of 3D GPR processing. However, the 2D GPR dramatically suffers from poor lateral resolution when lines are spaced far apart (beyond the horizontal resolution). The 3D GPR processing provides significant imaging improvements over conventional 2D GPR by providing an increase in lateral resolution from 17 m to 1.7 m in a glaciological setting with 25 MHz GPR antennas. Such an imaging improvement can be seen by comparing the extracted englacial conduit reflection from a sparse network of 2D GPR profiles in 2019 as described in Church et al. (2020) and the 3D GPR processing described here in 2020. The extracted amplitude of the englacial conduit (Fig. S3) from the 3D GPR processing in the north of the survey (section A in Fig. 4a) shows conduit width of $8 \pm 1.7$ m; whereas the 2D GPR suggest a significantly wider englacial network $17.5 \pm 8.5$ m.

The major limiting factors of such future 3D GPR surveys are the time-consuming nature of ground-based GPR data acquisition, the accessibility of the field site due to dangers on the glacier such as heavily crevassed areas and the safety of personnel carrying heavy GPR systems. All of these concerns could be addressed with drone technology. Drone technology is often used in cryosphere research (Gaffey and Bhardwaj, 2020) however GPR-based drone surveys are currently limited to landmine detection (Colorado et al., 2017; Sipos et al., 2017) and soil mapping (Wu et al., 2019). Developments of lightweight GPR systems are anticipated to provide the possibility of equipping small, uncrewed aerial vehicles with the capabilities to acquire 3D radar data in a fast and efficient manner. With sufficient power a drone-based solution would acquire the Rhonegletscher data within an estimated 7 hours of flying time, instead of 9 days spent for the ground-based study.

## 6 Conclusions

By using a 3D GPR data set, we have produced unaliased imaging of the Rhonegletscher's drainage network in a section of its ablation zone that has led to the agreement of long-standing glacier hydraulic theory. Upon meeting an overdeepening, melt water is routed alongside the flank of the glacier within so-called gradient conduits and thereby avoiding the overdeepening.

The geometry of the drainage network was determined by extracting the root-mean-squared reflected GPR amplitude. Using this extracted GPR attribute, we were able to delineate a hydrological system in 3D, which includes connected englacial and subglacial conduits. Such observations were only possible due to the 3D nature of our data. 2D GPR imaging would have failed in determining the continuity of this hydraulic system and with such 2D GPR data a connection would only be the result of speculation. We found the dimensions of the conduit were 60 times wider than its thickness, which is in contrast to theory that conduits are circular. However, these observations are in line with further conduit geometry developments by Hooke et al.

(1990). From the geometry of the conduit network, we are able to confirm that the hydraulic system is an efficient drainage network.

In addition to observing the main efficient drainage network, the extracted GPR reflected amplitude from the glacier's basal interface suggested that subglacial water is potentially pooling in numerous isolated localised flat areas. This localised pooling of water forms an inefficient drainage network. Thereby, both an efficient and inefficient drainage networks are able to coexist within overdeepenings.

3D GPR data have been adopted and have proven to be successful for imaging small-scale targets within the fields of archaeology and investigating shallow fault zones, and to a lesser extent in glaciological investigations. This study illustrates the feasibility and the opportunities that are offered by implementing 3D GPR to image glaciers and their hydraulic networks. Alongside the development of lightweight GPR systems and uncrewed aerial vehicles, such future 3D GPR surveys will be acquired faster and in a more efficient manner and thereby ultimately lead to significant improvements in our understanding of glacier hydrology.

*Video supplement.* Movies showing inline, cross and depth slices through the 3D GPR cube can be found at https://doi.org/10.3929/ethz-b-000471304

*Author contributions.* GC, MG, AB, and HM designed the 3D GPR experiment, which were carried out by GC. GC processed the data and interpreted the data, with help from all co-authors. GC wrote the manuscript with comments and suggestions for improvements from all co-authors.

*Competing interests.* The authors declare that they have no conflict of interest.

*Financial support.* This research has been supported by the Schweizerischer Nationalfonds zur Förderung der Wissenschaftlichen Forschung (Grant no. 200021_169329/1 and 200021_169329/2).

*Acknowledgements.* Data acquisition has been provided by the Exploration and Environment Geophysics (EEG) group and the Laboratory of Hydraulics, Hydrology and Glaciology (VAW) of ETH Zurich. We especially thank L. Bührle, E. Mattea, D. May, V. Krampe and C. Walker for their support during the field data acquisition. We would like to gratefully acknowledge the Landmark Graphics Corporation for providing data processing software through the Landmark University Grant Program. The authors wish to acknowledge all volunteers for their valuable help in the fieldwork. Furthermore, we would like to thank both Daniel Farinotti for an insightful in-house review, which improved the clarity

of the manuscript and Mauro Werder for important insights into the discussion of the paper. We would also like to thank the editor Joseph

330    MacGregor and the two anonymous reviewers for their constructive comments in order to improve the quality of the manuscript.

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
