# Peer review of "Ground-penetrating radar imaging reveals glacier's drainage network in 3D"

_The Cryosphere, 2021_

## Referee Comment (RC1)

**Comment on "Ground-penetrating radar imaging reveals glacier's drainage network in 3D"**

12 May 2021

**Summary**

The manuscript presents a 3D ground-penetrating radar (GPR) survey near the termini of Rhonegletscher, Switzerland. The goals of the study are to characterize the englacial/subglacial drainage system, as well as highlight the advantages of 3D GPR over glaciers, which this study is the first of its kind. Based on manual inspection of the radargrams an englacial drainage network is outlined with high resolution. The authors then use the amplitude of the picked network reflection, as well as the bed return to derive locations of englacial/subglacial water. The results are in agreement with two main theoretical frameworks - i) the englacial drainage network leads around an overdeepening rather than water flowing directly across it, and ii) the drainage conduit is likely flat and non-circular shaped.

**General minor comments**

According to the authors-, and my knowledge, this is the first 3D GPR study over a glacier, resulting in detailed characterization of the Rhonegletscher's englacial/subglacial hydrological drainage system. The methodologies and results are mostly well explained, and the manuscript is well structured, while the writing could be improved with some minor changes. I believe that the observations bring a valuable contribution to the glaciology and radioglaciology community and is well within the scope of The Cryosphere. However, I find that the main weak point of the manuscript is the lack of a discussion of possible implications of the findings for glacier dynamics and hydrology, especially as this is the motivation for studying such drainage networks (as outlined in the manuscripts introduction). Additionally, the manuscript could be improved in several aspects outlined below:

**Introduction**: The introduction provides a decent overview of the topic, stating the importance of a glaciers subglacial hydrological system for mass loss, and the difficulties/lack of observations using 2D GPR measurements. However, there are a few minor points that could be improved:

1) The differences between 2D and 3D GPR is not entirely clear and could be introduced in a bit more detail (i.e. both datasets are collected as line-by-line surveys, but with different survey setups/line spacing and processing techniques)

2) I think that the research goals could be formulated more clearly and more closely tied to the current knowledge gaps. For example, the manuscript mentions the study is motivated by previous findings (Line 57). I therefore suggest including a brief overview of these previous observations regarding the glacier hydrology on the Rhonegletscher, and what aspects of these findings (i.e. unknown extent of drainage system, improved resolution?) motivated this study. A brief introduction to the current knowledge on the Rhonegletscher drainage system would also help the reader to better understand the results from this study.

3) The introduction would greatly benefit from a brief overview of the theory of englacial drainage flow around overdeepenings, rather than simply mentioning it as a "long-standing glacier hydraulic theory (L45)".

**Methods:** The description of the methods is generally good, but I think the interpretation of the drainage network would benefit from a bit more detailed explanation on how the drainage network was picked (i.e. manual picking of visually strong, coherent reflections, added knowledge from previous GPR/seismic/boreholes data (Church et al., 2019, 2020)). I find it particularly difficult to follow how the drainage network was identified at the glacier bed. For example, on Figure 2c there is no obvious visual difference between the picked subglacial drainage network (dark blue arrows) and the ice-bed interface at profile locations 1850 and 1900. Additionally, in Section 4.3, the manuscript states that high basal reflectivity regions may also represent a subglacial water/drainage system, but these were not identified as drainage network in the manual picking. This leaves me wondering why these areas were not picked in the first place, and whether the identification of *subglacial water* should be done via interpreting bed amplitudes rather than visually identifying/picking a drainage network.

Additionally, Church et al., (2020) note the importance of calibrating reflectivities using borehole data. Was this attempted in this study? Or could it be done via comparing the 3D grid from this study to the previous GPR data?

**Results:** In general, the results are well described, however, there are two main points that could improve the manuscript:

1) Rather than stating that water is pooling where the bed is flat, it should be stated in terms of subglacial hydraulic flatness. I suggest calculating the subglacial hydraulic head/gradient, and determine whether the areas of high basal reflectivity occur in local hydraulic minima (which would allow pooling). Additionally, I wonder whether the high bed reflectivities could be caused by saturated sediments (or clay, see (Tulaczyk & Foley, 2020)) rather than ponded water.
2) The comparison of the 2D vs. 3D processing is based on a single profile. Because there is so much data available, I think the argument for 3D processing would be stronger if more data is shown (i.e. more profiles in a supplement, and/or statistics showing the difference between bed/englacial reflections identified in the 2D vs. 3D data). I also think that there would be an opportunity to compare results from this study's 3D data to the (already picked) 2D data in Church et al., (2020). Considering the manuscripts goal "to demonstrate the feasibility and opportunities of 3D GPR", I think that a more sophisticated comparison between 2D and 3D data would better highlight the advantages of 3D GPR.

**Discussion**: Overall, the discussion is well structured, however, is a bit vague regarding some findings:

1) I think the manuscript would benefit from a more detailed discussion on the configurations of the drainage system and its implications for glacier hydraulics, possible seasonal evolution and ice dynamics. For example, the results show a channelized water system

upstream, and a more dispersed system downstream. What causes this and what are the implications for ice dynamics? What are the implications of the englacial drainage network connecting to the basal drainage network and vice versa? And what implications could be derived from the study's results regarding future glacier retreat and the formation of the proglacial lake?

2) The results appear to agree with the theory of non-circular channel shapes, however, there is no discussion about the implications of this agreement. What does this mean for the water pressure, channel evolution, ice dynamics?

3) Finally, the manuscript would benefit from a discussion of the findings with respect to previous observations on the Rhonegletscher. For example, does the 3D data agree with the previous 2D survey in the upstream part? What knowledge is gained from the 3D survey compared to the 2D survey from Church et al. (2020)?

**Figures/Movies:** Most figures are very well prepared and are easy to read/understand. I also highly appreciate the movies which helped to understand the results. I have a few small suggestions for the figures:

Figure 2b: Maybe I am seeing this wrong, but it looks like the bed contour lines are different than in the following figures. It appears that the overdeepening minima is west of profile line C, whereas in the following figures the overdeepening minima is east of profile line C.

Figure 4a: The dark blue outline in conduit region C is difficult to see, I suggest using a different color.

Figure 5: I wonder if it would be beneficial to mark the hydraulic head contour lines instead of the bed contour lines. Additionally, it might be useful to mark the outline of the picked drainage network to highlight the difference in englacial/subglacial water pathways.

**Line-by-line minor comments:**

L33-34: "2D data sets are typically unable to image complex subsurface structures, …" I'm not sure if it is the *complexity* of subsurface structures that is difficult to image with 2D radar datasets, or rather the *size/scale* of subsurface structures relative to the radar surveys.

L36: Suggest replacing "option" with "tool"

L39: "…, because 3D GPR provides subsurface images that can be viewed from arbitrary directions,…" I agree that being able to view/visually inspect subsurface images in 3D provides an advantage, but results from 2D survey grids could be interpolated to generate (lower resolution) 3D images as well. I believe that the main advantage, and thus the argument for 3D surveys (or simply closely-spaced survey grids) is the high data coverage allowing to image the target at high spatial resolution. The "high resolution" is also noted as the motivating factor on Line 46. I suggest adding the high resolution aspect to this sentence.

L43-46: I think this paragraph would benefit from some re-structuring. It is not clear whether the 3D GPR survey was performed to demonstrate the feasibility of such surveys, to further characterize the drainage network previously identified in (Church et al., 2019, 2020), or to

investigate the hypothesis that englacial drainage flows around overdeepenings (or all of the above).

L45: "… to confirm long-standing glacier hydraulic theory." I think confirm is a strong word, and I am not sure if a theory can be confirmed with just one observation. I suggest changing this to "our hydrological observations are in agreement with a long-standing glacier hydraulic theory". Additionally, I think it would be helpful to spell out what the theory is.

L49: Replace 'It is representative…" with "The Rhonegletscher is representative…"

L70-71: "The 3D GPR data were collected …" I'm a bit hesitant to call this 3D GPR data, as the data was collected along 2D profile lines, just with close line spacing. Maybe this could be specified by something along "The GPR data for our 3D processing flow were collected in dense (2m spaced) survey lines perpendicular to the ice flow direction."

L93: "performed using an EM wave propagation velocity" (insert an)

L93: I suggest replacing "stretched" with "converted"

L95: I am not familiar with Q compensation for attenuation, but is it possible to state the attenuation rates used in the study (typically expressed as dB/km)? And what are the uncertainties from this attenuation correction?

L99: "…, the drainage network was picked …", it is unclear on what basis the drainage network was identified, i.e. manual picking of visually strong, coherent reflections, added knowledge from previous GPR/seismic/boreholes data (Church et al., 2019, 2020)?

L121-123: The sentences are a bit longwinded and repetitive. Also, rather than just stating that the drainage network was identified from the GPR data, I suggest clarifying based on which GPR attributes the network was interpreted from (i.e. based on the high amplitudes, manual inspection, the spatial pattern, the agreement with previous observations, see comment above). Additionally, how are the low amplitudes towards the edges and southern part of the outlined drainage network interpreted? Are these areas of past water flow, channel filled with air/sediments?

L123: "red" should be "yellow"

L125&L126: replace "overdeepen" with "overdeepening"

L126: "flowing alongside", a conduit doesn't really flow itself, replace with "follows" or "runs"

L128/130/134: "flows into a subglacial drainage system" / "flows into another englacial conduit", same as above, I suggest changing this to "connects with", or "transitions to …"

L132: replace "the conduit is expected to flow" with "water in the conduit is expected to flow"

L145: add "local basal hydrological conditions" to be more specific

L145-147: "… thereby indicating this area is positioned along the main drainage network identified in Figure 4.". The message of this sentence is not clear. What does "along the main drainage network" mean? I assume the argument is that the high reflectivities suggest the presence of water

at the glacier bed, with the upstream and downstream boundaries of this area spatially coinciding with the location of the englacial drainage network identified in Figure 4.

L148: Delete s in "amplitudes"

L154: Delete "today's"

L155: In my opinion, 2D surveys can provide 3D subsurface images via interpolation, but the distinct advantage of 3D surveys is the image resolution. I suggest changing to "unable to provide high resolution 3D subsurface images, …"

L157-161: I think "ambiguities" or "off-nadir reflections" would be a better description than "distortions".

L158: Figure 6 refers to an example geometry that can lead to off-nadir reflections in 2D GPR surveys, but it does not show the "distortion"/ambiguity itself.

L163: Delete "improved"

L167: I suggest changing "more unambiguous" to "less ambiguous"

L174: "In our case, it has a meandering nature …" I suggest changing this sentence to: "The Rhonegletscher drainage network identified in this study has a meandering nature throughout the survey area, with an increasing network width towards the glacier terminus."

L182-183: "The 3D GPR imaging results…" this sentence is a repetition from above (L175), I suggest merging them.

L185: I think it would be great to include a bit more detail on how exactly the Rhonegletscher results are in agreement with Hooke et al. (1990).

L194-197: There is no figure in the results section that shows the hydraulic potential/gradient. I suggest adding hydraulic potential contour lines to one of the maps (e.g. on Figure 5a, replacing bed elevation contours). This would also take care of the argument about high reflectivity in areas where water has the potential to pool. Alternatively, I suggest adding a figure to show the hydraulic head/gradient (possibly as supplement).

L202: "This is in contrast to our 3D GPR data set,…" I suggest removing or re-wording this sentence, as the GPR dataset in the study is not over a subglacial lake, but the sentence refers to the delineation of subglacial lake outlines.

L207-210: Here, the water accumulations are interpreted as isolated cavities, but in the results section (L150), it is noted that in the southern area there is likely a connected water system. If there is a different interpretation of the hydraulic system in the north and south, this should be stated more clearly.

L218: Not clear what is meant by "rate of acquisition". Time-consuming dense survey grids required for 3D surveys?

L218: I am not sure if the "accessibility" of the field site is more difficult for 3D surveys than 2D surveys. When conducting a 2D profile across an ice cap/glacier, I would expect the glacier to be

similarly accessible a few meters upstream/downstream of this profile line (with the exception of heavily crevassed areas). I think another argument for UAVs would be safety (i.e. less time spent on the glacier, no need to cross crevasses etc.).

L227: "… confirming long-standing…", in my opinion, confirming is a strong word here (see comment above).

L233-234: "… which is in contrast to theory. However, these observations are in line with further conduit geometry developments…". This is a bit vague, I suggest clarifying and spelling out that the results agree with the theory of broad and low shaped channels rather than circular channels.

L235: delete "as"

**References**

Church, G., Bauder, A., Grab, M., Rabenstein, L., Singh, S., & Maurer, H. (2019). Detecting and characterising an englacial conduit network within a temperate Swiss glacier using active seismic, ground penetrating radar and borehole analysis. *Annals of Glaciology*, *60*(79), 193–205. https://doi.org/10.1017/aog.2019.19

Church, G., Grab, M., Schmelzbach, C., Bauder, A., & Maurer, H. (2020). Monitoring the seasonal changes of an englacial conduit network using repeated ground-penetrating radar measurements. *Cryosphere*, *14*(10), 3269–3286. https://doi.org/10.5194/tc-14-3269-2020

Tulaczyk, S. M., & Foley, N. T. (2020). The role of electrical conductivity in radar wave reflection from glacier beds. *Cryosphere*, *14*(12), 4495–4506. https://doi.org/10.5194/tc-14-4495-2020

---

## Author Comment (AC1)

Dear Dr. Joseph MacGregor and Reviewers,

We appreciate all your positive and constructive feedback to our manuscript tc-2021-82 entitled "Ground-penetrating radar imaging reveals glacier's drainage network in 3D". On the following pages, we have provided a point-by-point response to your comments

If you have any further questions, we would happily answer them and we look forward to hearing back from you regarding your decision.

Best regards, Gregory Church and all co-authors

**Review 1**

**Comment on "Ground-penetrating radar imaging reveals glacier's drainage network in 3D" 12 May 2021**

**Summary**

The manuscript presents a 3D ground-penetrating radar (GPR) survey near the termini of Rhonegletscher, Switzerland. The goals of the study are to characterize the englacial/subglacial drainage system, as well as highlight the advantages of 3D GPR over glaciers, which this study is the first of its kind. Based on manual inspection of the radargrams an englacial drainage network is outlined with high resolution. The authors then use the amplitude of the picked network reflection, as well as the bed return to derive locations of englacial/subglacial water. The results are in agreement with two main theoretical frameworks - i) the englacial drainage network leads around an overdeepening rather than water flowing directly across it, and ii) the drainage conduit is likely flat and non-circular shaped.

We are grateful for the reviewer's positive and constructive feedback and appreciate the suggestions to help further improving our manuscript.

**General minor comments**

According to the authors-, and my knowledge, this is the first 3D GPR study over a glacier, resulting in detailed characterization of the Rhonegletscher's englacial/subglacial hydrological drainage system. The methodologies and results are mostly well explained, and the manuscript is well structured, while the writing could be improved with some minor changes. I believe that the observations bring a valuable contribution to the glaciology and radioglaciology community and is well within the scope of The Cryosphere. However, I find that the main weak point of the manuscript is the lack of a discussion of possible implications of the findings for glacier dynamics and hydrology, especially as this is the motivation for studying such drainage networks (as outlined in the manuscripts introduction). Additionally, the manuscript could be improved in several aspects outlined below:

**Introduction**: The introduction provides a decent overview of the topic, stating the importance of a glaciers subglacial hydrological system for mass loss, and the difficulties/lack of observations using 2D GPR measurements. However, there are a few minor points that could be improved:

1) The differences between 2D and 3D GPR is not entirely clear and could be introduced in a bit more detail (i.e. both datasets are collected as line-by-line surveys, but with different survey setups/line spacing and processing techniques) – We have added a sentence within the introduction stating how 3D GPR surveys are composed of densely spaced multiple 2D GPR profiles that are collectively processed in order to avoid both any sampling bias in the in-line direction and spatial aliasing in the cross-line direction.

2) I think that the research goals could be formulated more clearly and more closely tied to the current knowledge gaps. For example, the manuscript mentions the study is motivated

by previous findings (Line 57). I therefore suggest including a brief overview of these previous observations regarding the glacier hydrology on the Rhonegletscher, and what aspects of these findings (i.e. unknown extent of drainage system, improved resolution?) motivated this study. A brief introduction to the current knowledge on the Rhonegletscher drainage system would also help the reader to better understand the results from this study. – We have renamed 'Survey Site' to be 'Survey Site and Previous Work'. We have moved and expanded on the previous GPR surveys conducted on Rhonegletscher as suggested.

3) The introduction would greatly benefit from a brief overview of the theory of englacial drainage flow around overdeepenings, rather than simply mentioning it as a "long-standing glacier hydraulic theory (L45)". – We have added a paragraph before the aim of the study and stated how the overdeepening hypothesis from Lliboutry (1983) can have an impact on sliding velocities.

**Methods**: The description of the methods is generally good, but I think the interpretation of the drainage network would benefit from a bit more detailed explanation on how the drainage network was picked (i.e. manual picking of visually strong, coherent reflections, added knowledge from previous GPR/seismic/boreholes data (Church et al., 2019, 2020)). I find it particularly difficult to follow how the drainage network was identified at the glacier bed. For example, on Figure 2c there is no obvious visual difference between the picked subglacial drainage network (dark blue arrows) and the ice-bed interface at profile locations 1850 and 1900. Additionally, in Section 4.3, the manuscript states that high basal reflectivity regions may also represent a subglacial water/drainage system, but these were not identified as drainage network in the manual picking. This leaves me wondering why these areas were not picked in the first place, and whether the identification of subglacial water should be done via interpreting bed amplitudes rather than visually identifying/picking a drainage network. -We have updated the manuscript within section 3.2 with more details on how the drainage network was picked by visually following the strongest continuous reflection across the survey site. This was quality controlled using an amplitude map (RMS map) between the ice-bed interface and the glacier surface. In Figure 2c, the strong bed reflection identified at profile location 1850 and 1900 is not connected to the continuous reflection and therefore, these have not been interpretated along the main drainage network. The areas that were not picked in Section 4.3 were not part of the main drainage network (i.e., not showing similar characteristics to the drainage network – amplitude strength/continuality etc). We have updated Section 4.3 to state this.

Additionally, Church et al., (2020) note the importance of calibrating reflectivities using borehole data. Was this attempted in this study? Or could it be done via comparing the 3D grid from this study to the previous GPR data? – This was not attempted during this study as reflectivities were not calculated and instead only the reflected GPR amplitudes were plotted after amplitude correction (geometrical spreading and attenuation). In order to calibrate using the borehole information the amplitudes could be convert to reflectivities using the methodology described in Church et al (2020). However, this was beyond the scope of this project and could be a natural extension of this work.

**Results**: In general, the results are well described, however, there are two main points that could improve the manuscript:

1) Rather than stating that water is pooling where the bed is flat, it should be stated in terms of subglacial hydraulic flatness. I suggest calculating the subglacial hydraulic head/gradient, and determine whether the areas of high basal reflectivity occur in local hydraulic minima (which would allow pooling). Additionally, I wonder whether the high bed reflectivities could be caused by saturated sediments (or clay, see (Tulaczyk & Foley, 2020)) rather than ponded water. – We have added 4 different hydraulic potential scenarios figures in the supplement reflecting the diurnal nature of the subglacial water pressure of Rhonegletscher (25%, 50%, 75% and 95% water pressure). Furthermore, during the discussion we have added that high bed reflectivities may result from either water accumulations along a hard bed or saturated sediments. However, the latter is unlikely due to outcrops showing a granite bedrock with little sediment and also borehole camera images from 2018 within the survey indicating a hard granite basal-interface.

2) The comparison of the 2D vs. 3D processing is based on a single profile. Because there is so much data available, I think the argument for 3D processing would be stronger if more data is shown (i.e. more profiles in a supplement, and/or statistics showing the difference between bed/englacial reflections identified in the 2D vs. 3D data). I also think that there would be an opportunity to compare results from this study's 3D data to the (already picked) 2D data in Church et al., (2020). Considering the manuscripts goal "to demonstrate the feasibility and opportunities of 3D GPR", I think that a more sophisticated comparison between 2D and 3D data would better highlight the advantages of 3D GPR. – We have added a supplement of 3 additional 2D v. 3D GPR comparisons. We conclude that the imaging is similar in terms of basal reflectivity and englacial reflectivity between 3D and 2D however, the real impact is when observing multiple 2D lines (making a pseudo-2.5D survey) and the 3D cube. In such a case there is a visible difference (see discussion part 3 below for the comparison between multiple 2D lines and 3D imaging).

**Discussion**: Overall, the discussion is well structured, however, is a bit vague regarding some findings:

1) I think the manuscript would benefit from a more detailed discussion on the configurations of the drainage system and its implications for glacier hydraulics, possible seasonal evolution and ice dynamics. For example, the results show a channelized water system upstream, and a more dispersed system downstream. What causes this and what are the implications for ice dynamics? What are the implications of the englacial drainage network connecting to the basal drainage network and vice versa? And what implications could be derived from the study's results regarding future glacier retreat and the formation of the proglacial lake? – Regarding glacier hydraulics and glacier ice dynamics we have addressed this in point 2 point below. In terms of seasonal evolution without having repeated surveys throughout the year any such statements regarding the subglacial drainage network are speculative and difficult to make. We know from previous measurements (Church et al. 2020) that the main network does not completely shut during winter however, we do not have continuous surface ice flow velocities to tie with our study and this would also be a nature extension to combine more data.

2) The results appear to agree with the theory of non-circular channel shapes, however, there is no discussion about the implications of this agreement. What does this mean for the water pressure, channel evolution, ice dynamics? – We have updated section 5.2 stating that in theory Hooke-channels lead to increased hydraulic friction and thus higher water pressure than semi-circular R-channels. This is due to not only the shape itself but also due to higher closure rates. Thus, the impact on ice dynamics is then that such a configuration would support higher sliding speeds. Furthermore, we now refer to Werder et al 2010 who fitted a Hooke-channel to tracer measurements and found that the hydraulic friction could be well explained by assuming low and broad channels (i.e., Hooke channels).

3) Finally, the manuscript would benefit from a discussion of the findings with respect to previous observations on the Rhonegletscher. For example, does the 3D data agree with the previous 2D survey in the upstream part? What knowledge is gained from the 3D survey compared to the 2D survey from Church et al. (2020)? – We have included an additional figure of the drainage network derived (added to supplement) from the 2D GPR data presented in Church et al. 2020 and compared it to the 3D GPR drainage network. The 3D GPR data processing has improved the lateral resolution of the drainage network. We have updated the discussion section (5.3) on the future of 3D GPR within glaciology to discuss this uplift in lateral resolution and thereby advocating future 3D GPR surveys.

**Figures/Movies**: Most figures are very well prepared and are easy to read/understand. I also highly appreciate the movies which helped to understand the results. I have a few small suggestions for the figures:

Figure 2b: Maybe I am seeing this wrong, but it looks like the bed contour lines are different than in the following figures. It appears that the overdeepening minima is west of profile line C, whereas in the following figures the overdeepening minima is east of profile line C. – I assume you mean Fig. 1b – if so, this is correct. The contour lines in Fig 1b are from the GlaTe modelling and are not picked on using a 3D GPR processing dataset. The GlaTe model is an ice-thickness model that combines ice dynamics and GPR data to approximate ice-thickness. The caption has been updated to reflect this in Fig 1. The bed elevation in Fig 4 onwards appear only within the survey are. We have updated caption 4 and 5 to indicate that the basal elevation is from the 3D GPR data.

Figure 4a: The dark blue outline in conduit region C is difficult to see, I suggest using a different color. – We have updated the outline of the subglacial network zone to be a red colour.

Figure 5: I wonder if it would be beneficial to mark the hydraulic head contour lines instead of the bed contour lines. Additionally, it might be useful to mark the outline of the picked drainage network to highlight the difference in englacial/subglacial water pathways. – See note above regarding adding figures within the supplement showing hydraulic potential.

Line-by-line minor comments:

L33-34: "2D data sets are typically unable to image complex subsurface structures, ..." I'm not sure if it is the complexity of subsurface structures that is difficult to image with 2D radar datasets, or rather the size/scale of subsurface structures relative to the radar surveys. – We

believe that both the complexity of the subsurface (i.e., reflections from out of the 2D plane) and size/scale of subsurface structure would cause imaging issues. However, out-ofplane reflections would result in mis-positioning of reflection data within the plane and this can only be solved with 3D. For this reason, we have retained the original wording but have updated the manuscript to state both the complexity and scale of subsurface structures relative to the radar surveys make imaging difficult with 2D GPR data.

L36: Suggest replacing "option" with "tool" – Changed as requested

L39: "..., because 3D GPR provides subsurface images that can be viewed from arbitrary directions,..." I agree that being able to view/visually inspect subsurface images in 3D provides an advantage, but results from 2D survey grids could be interpolated to generate (lower resolution) 3D images as well. I believe that the main advantage, and thus the argument for 3D surveys (or simply closely-spaced survey grids) is the high data coverage allowing to image the target at high spatial resolution. The "high resolution" is also noted as the motivating factor on Line 46. I suggest adding the high-resolution aspect to this sentence. – We have added a new sentence stating that 3D GPR can provide high spatial resolution imaging of glacier drainage networks.

L43-46: I think this paragraph would benefit from some re-structuring. It is not clear whether the 3D GPR survey was performed to demonstrate the feasibility of such surveys, to further characterize the drainage network previously identified in (Church et al., 2019, 2020), or to investigate the hypothesis that englacial drainage flows around overdeepenings (or all of the above). – Done (all of the above), we have re-structured the sentence to have three list items stating that these are all the objectives.

L45: "... to confirm long-standing glacier hydraulic theory." I think confirm is a strong word, and I am not sure if a theory can be confirmed with just one observation. I suggest changing this to "our hydrological observations are in agreement with a long-standing glacier hydraulic theory". Additionally, I think it would be helpful to spell out what the theory is. – Done, rephrased confirm to determine whether the observations are in agreement with long-standing theory.

L49: Replace 'It is representative..." with "The Rhonegletscher is representative..." – Done

L70-71: "The 3D GPR data were collected ..." I'm a bit hesitant to call this 3D GPR data, as the data was collected along 2D profile lines, just with close line spacing. Maybe this could be specified by something along "The GPR data for our 3D processing flow were collected in dense (2m spaced) survey lines perpendicular to the ice flow direction." – Agreed, and modified text

L93: "performed using an EM wave propagation velocity" (insert an) - Modified

L93: I suggest replacing "stretched" with "converted" - Modified

L95: I am not familiar with Q compensation for attenuation, but is it possible to state the attenuation rates used in the study (typically expressed as dB/km)? And what are the

uncertainties from this attenuation correction? – More information on the attenuation correction has been addressed with reviewer 2 comments. For more details on the attenuation correction see Irving and Knight (2003).

L99: "..., the drainage network was picked ...", it is unclear on what basis the drainage network was identified, i.e. manual picking of visually strong, coherent reflections, added knowledge from previous GPR/seismic/boreholes data (Church et al., 2019, 2020)? – The drainage network was identified as the strongest continuous coherent reflection across the survey site and manually picked with aid from previous GPR, seismic and borehole studies. The section in the manuscript has been updated to reflect these changes.

L121-123: The sentences are a bit longwinded and repetitive. Also, rather than just stating that the drainage network was identified from the GPR data, I suggest clarifying based on which GPR attributes the network was interpreted from (i.e. based on the high amplitudes, manual inspection, the spatial pattern, the agreement with previous observations, see comment above). Additionally, how are the low amplitudes towards the edges and southern part of the outlined drainage network interpreted? Are these areas of past water flow, channel filled with air/sediments? – Alongside the comment above we have updated the processing section to include how the drainage network was identified and picked. This has been reflected in the revised manuscript. Furthermore, areas on the edges of the survey are not fully imaged and caution was used in these areas and there exist amplitude uncertainties within these areas.

L123: "red" should be "yellow" - Replaced

L125&L126: replace "overdeepen" with "overdeepening" – Agree and replaced.

L126: "flowing alongside", a conduit doesn't really flow itself, replace with "follows" or "runs" – Corrected as suggested

L128/130/134: "flows into a subglacial drainage system" / "flows into another englacial conduit", same as above, I suggest changing this to "connects with", or "transitions to ...". – Updated manuscript to reflect these changes.

L132: replace "the conduit is expected to flow" with "water in the conduit is expected to flow" – Updated manuscript with this change.

L145: add "local basal hydrological conditions" to be more specific. – **Updated manuscript to** state that the amplitude provides insights into the bedrock type and whether subglacial water is present.

L145-147: "... thereby indicating this area is positioned along the main drainage network identified in Figure 4.". The message of this sentence is not clear. What does "along the main drainage network" mean? I assume the argument is that the high reflectivities suggest the presence of water at the glacier bed, with the upstream and downstream boundaries of this area spatially coinciding with the location of the englacial drainage network identified in

Figure 4. – We have reworded this to state – thereby indicating this area is identical to the drainage network identified in Fig4a.

L148: Delete s in "amplitudes" - Done

L154: Delete "today's" - Done

L155: In my opinion, 2D surveys can provide 3D subsurface images via interpolation, but the distinct advantage of 3D surveys is the image resolution. I suggest changing to "unable to provide high resolution 3D subsurface images, …" – **Updated sentence as requested.**

L157-161: I think "ambiguities" or "off-nadir reflections" would be a better description than "distortions". – Retained distortions however, have added that the ray path in Figure 6 results in distortions caused by off-nadir reflections.

L158: Figure 6 refers to an example geometry that can lead to off-nadir reflections in 2D GPR surveys, but it does not show the "distortion"/ambiguity itself. – We have altered the caption to state the example shows an off-nadir reflection raypath that would lead to distortions in 2D GPR processing.

L163: Delete "improved" - Deleted

L167: I suggest changing "more unambiguous" to "less ambiguous" – Changed as suggested

L174: "In our case, it has a meandering nature ..." I suggest changing this sentence to: "The Rhonegletscher drainage network identified in this study has a meandering nature throughout the survey area, with an increasing network width towards the glacier terminus." – Updated as suggested

L182-183: "The 3D GPR imaging results..." this sentence is a repetition from above (L175), I suggest merging them. – These sentences are now combined in order to avoid repetition.

L185: I think it would be great to include a bit more detail on how exactly the Rhonegletscher results are in agreement with Hooke et al. (1990). – We have added to the manuscript that Hooke used water pressure data from Storglaciären to establish a theory that subglacial channels are broad and low and therefore, are in agreement to our observations on Rhonegletscher.

L194-197: There is no figure in the results section that shows the hydraulic potential/gradient. I suggest adding hydraulic potential contour lines to one of the maps (e.g. on Figure 5a, replacing bed elevation contours). This would also take care of the argument about high reflectivity in areas where water has the potential to pool. Alternatively, I suggest adding a figure to show the hydraulic head/gradient (possibly as supplement). – We have added supplement figures of the hydraulic head. In order to calculate the hydraulic potential, the subglacial water pressure is required. We have generated several hydraulic potential plots with subglacial water pressure varying between 25% and 95% of the overburden ice thickness (in agreement with the diurnal fluctuations observed in Sugiyama 2007). The

**overall conclusion that the isolated high amplitude spots lie in places where the hydraulic potential is constant.**

L202: "This is in contrast to our 3D GPR data set,..." I suggest removing or re-wording this sentence, as the GPR dataset in the study is not over a subglacial lake, but the sentence refers to the delineation of subglacial lake outlines. – We have reworded this sentence to state "With the user of our 3D GPR data set, we are able to delineate high-resolution changes to the basal interface".

L207-210: Here, the water accumulations are interpreted as isolated cavities, but in the results section (L150), it is noted that in the southern area there is likely a connected water system. If there is a different interpretation of the hydraulic system in the north and south, this should be stated more clearly. – This is corrected. In the results section we originally stated that in the southern part, high amplitude patches are connected. We have modified to state partially connected.

L218: Not clear what is meant by "rate of acquisition". Time-consuming dense survey grids required for 3D surveys? – Indeed, in order to avoid confusion with future readers we have updated the manuscript to state the major limiting factors are the time-consuming nature of the ground-based GPR data acquisition.

L218: I am not sure if the "accessibility" of the field site is more difficult for 3D surveys than 2D surveys. When conducting a 2D profile across an ice cap/glacier, I would expect the glacier to be similarly accessible a few meters upstream/downstream of this profile line (with the exception of heavily crevassed areas). I think another argument for UAVs would be safety (i.e. less time spent on the glacier, no need to cross crevasses etc.). – In terms of accessibility, the presented 3D GPR survey could have been extended in the west section. However, due to heavily crevassed areas (accessibility issues) this was not possible. We have updated the manuscript to state both the accessibility issues in terms of crevassed areas and also safety.

L227: "... confirming long-standing...", in my opinion, confirming is a strong word here (see comment above). – We have reworded accordingly.

L233-234: "... which is in contrast to theory. However, these observations are in line with further conduit geometry developments...". This is a bit vague, I suggest clarifying and spelling out that the results agree with the theory of broad and low shaped channels rather than circular channels. – Updated manuscript to reflect that the theory states channels are circular.

L235: delete "as" – Deleted.

**Reviewer 2**

**Review of "Ground-penetrating radar imaging reveals glacier's drainage network in 3D" by Gregory Church et al., May 2021**

**General Comments:**

This paper presents the results of a high-resolution, 3D ground-penetrating radar (GPR) experiment conducted near the terminus of the Rhonegletscher in Switzerland. Approximately ~85 line km of GPR data were acquired with 25-MHz antennas along a series of parallel survey lines oriented perpendicular to glacier flow. A dense (2-m) line spacing was used in order to avoid spatial aliasing of reflection events in the cross-line direction. By examining the spatial distribution of reflection amplitudes in the processed 3D GPR data cube, the authors are able to clearly identify and map major englacial and subglacial channels, which allows them to importantly confirm that englacial conduits tend to flow around glacial overdeepenings rather than directly over them. Further, they identify a number of other high-amplitude zones near the glacier bed that may represent accumulations of subglacial water.

Overall, I found this paper to be of excellent quality and think that it represents a very interesting contribution to the existing literature. The amount of work to acquire these data (on foot!) is impressive, and the results strongly encourage the continued use of dense 3D GPR acquisitions in glacier hydrological studies. My suggested revisions (see below) are rather minor and mainly focus along the following three themes:

**We are grateful for the reviewer's positive and constructive feedback and appreciate the suggestions to help further improving our manuscript.**

1) The authors should further acknowledge previous work involving dense 3D GPR acquisitions on glaciers and avoid statements suggesting that this is the first study of this kind. The study is excellent and the findings are extremely interesting, but it is not the first time that people have considered these kinds of data, even within the context of glacier hydrology. – A new paragraph has been added to the introduction reviewing the other 3D GPR data used to investigate the drainage network (namely, Harper et al and Egli et al) and we have removed statements indicating that this is the first study doing this (from what we can find, this is the first 3D GPR for englacial drainage network).

2) The authors should reduce the conclusive nature of a number of statements in the manuscript concerning channel widths and heights, data resolution, and the presence of subglacial water. For me, many of these findings are not absolute and the corresponding uncertainty should be clearly expressed in the interpretation. We have updated the conclusive nature of our statements to include uncertainty. For the lateral dimensions we have used the post-migration horizontal resolution as uncertainties and for the conduit thickness, the uncertainties stem from a GPR modelling exercise taken directly from Church et al. 2020 that investigate thin layer effects on the reflectivities.

3) A more in-depth discussion of some aspects of the GPR data processing, as well as on resolution, should be provided. – We have altered the part about the regularisation and

**provided some more detailed feedback in the line-by-line comments (including updates to the lateral resolution).**

**Specific comments:**

Line 6: Please delete "for the first time" and "unprecedented" from this sentence in the abstract. As much as the results presented in this paper are truly excellent and impressive, the wording suggests that such data have never been acquired before. Harper et al. (2010) use high-resolution, unaliased 3D GPR data to identify basal crevasses forming part of the subglacial drainage system of Bench Glacier, Alaska. More recently, Egli et al. (2021) identify the subglacial channel network on two Swiss glaciers from unaliased 3D GPR surveys. Hansen et al. (2020) also use 3D GPR to map the englacial and subglacial drainage system in a High-Arctic glacier, albeit in this latter case the survey lines were spaced quite far apart. – As suggested, we have removed this wording as we are not completing this for the first time. It is noted that Haper et al., used their 3D GPR for basal crevassing which didn't providing imaging of a conduit-based drainage network however, the recent Egli et al. manuscript had similar observations to ours.

Lines 27-46: The introduction of the paper is quite good, but very short, and for me what is missing is a summary and acknowledgement of work where people have used similarly unaliased 3D GPR surveys to investigate glaciers. Papers to be mentioned specifically in this context include Harper et al. (2010), Murray and Booth (2010), Reinardy et al. (2019), and Egli et al. (2021). Of these, Harper et al. (2010) and Egli et al. (2021) specifically investigate the glacier drainage system. – Done, in agreement with reviewer 1; we have updated the introduction to include a more comprehensive review of 3D GPR literature, 2D v. 3D GPR, overview of research goals and overview of englacial drainage flow around overdeepenings.

Line 89, "Such a processing step...": This sentence is confusing. What do you mean by an amplitude imprint? – Regularisation is the processing stage used for 3D geophysical processing – typically done in large scale seismic processing for offshore hydrocarbon exploration and this has been applied in this GPR processing workflow. Regularisation is the process of moving the geophysical data to centre of their respective bins. Migration results improve when the GPR data has uniform spacing (i.e., constant 0.5 m in inline and 2 m spacing in crossline). The regularisation can be done using anti-leakage Fourier Transforms as described here: <a href="https://doi.org/10.1190/1.3507248">https://doi.org/10.1190/1.3507248</a> – fundamentally, it's an interpolation of the GPR data to its bin centre. If the bin centre interpolation was done without removing overfold within the bins an amplitude imprint of overfold could be observed. We have modified this section of the manuscript to explain it clearer.

Line 90: Please provide details on how exactly the data were "interpolated and regularized". What do you mean by regularization? – See above – I have removed the term regularisation to avoid confusion and replaced with interpolation of GPR data within their own bin.

Line 93: Please provide some further details on the Kirchhoff migration procedure. How did you choose the constant velocity of 0.167 m/ns, which corresponds to ice with essentially zero water content? And what aperture was used for the migration? I assume that no

corrections for the radiation pattern of the antennas were included (?) Did you account for the effects of the glacier surface topography?

We have updated the manuscript to reflect the additional processing information outlined below:

- Velocity; we used 0.167 m/ns based upon previous studies on temperate glaciers (Langhammer et al 2019). In Church et al 2020, CMP's were acquired to determine velocities and were found to be between 0.165 and 0.17 m/ns.
- Migrations were limited based upon dip rather than aperture. A dip limit of 50 degrees was used. This aided in the reduction of noise in the final image. Test were conducted from 30 degrees to 80 degrees.
- No correction for radiation pattern was performed. Amplitude corrections were uniform (i.e., no directionality correction) using a geometrical spreading and attenuation correction.
- Topographic correction was applied post-migration.

Line 95: More details on the Q compensation are needed. How significant was the dispersion in the data and why? – The attenuation correction was an amplitude-only correction. The dominant (amplitude) loss is through dielectric attenuation and therefore, Q-based attenuation correction was applied in order to correct for this loss.

Line 95: The data were already once bandpass filtered. Now, after migration, you mention that they were bandpass filtered again. Please explain why exactly this second filtering is necessary. – The amplitude Q compensation artificially boosted high frequencies that were noise and therefore, this noise was suppressed using the second bandpass filter. The reasoning has been added to the manuscript.

Line 100, "The spatial extent...": This step is not at all clear and needs further details and explanation. – We have re-worded the section to state; In order to ensure the picked drainage network encompassed the entire observable drainage network in the GPR survey area, GPR elevation slices were investigated in order to locate strong englacial and subglacial reflections that could represent a water-filled drainage network.

Line 110: You mention that a weak ice-bed reflection indicates that subglacial water is not present, but there could be other explanations. For example, the bed reflection may change amplitude as a result of variable success of the migration of the data. That is, assumptions in the migration (e.g., constant velocity no radiation patterns) along with the biased nature of the sampling (0.5 m in-line; 2 m cross-line) may cause amplitude artifacts along the bed. Also, I would think that, at a frequency of 25 MHz, you would need quite a thick layer of water at the bed to be seen (i.e., water may still be there, but in a thinner layer). – Agreed that the bed reflection may result of variable success of migration however, the migration velocity was tested (and alongside CMP's) and results within ±5% were similar. It would be expected that directional radiation patterns would provide amplitude imprints on strongly dipping reflectors due to the angle of incidence being away from 0 degrees. However, the glacier flank (strongest dipping event) GPR reflection produced a strong continuous reflector and thereby indicating that the radiation pattern is minimal on the reflected data. The largest amplitude differentiator would come from the actual subglacial/englacial material causing the reflection. In terms of thickness of water layer an often quote minimum thickness is

**$1/30^{th}$ wavelength (Sheriff et al.) and in the case of GPR wavelength in water this would equate to ~4 cm (wavelength at 25 MHz = 1.333 m). We find it difficult to believe that a thin water layer exists that is undetectable.**

Line 122, "The entire drainage network was identified from the GPR data": I think this sentence needs to be revised to reflect the fact that only the parts of the drainage network within the resolution limits of the GPR data were identified. There may be many more smaller englacial and subglacial conduits that were simply not detected with the 25-MHz data because of its rather low resolution. – We have updated the manuscript to state that "the entire detectable drainage network..."

Line 123, "red in Fig. 4a": There is no red in Figure 4a. – Changed to yellow; a late change of the colour map was done and this red to yellow was missed. Thank-you for bringing this to our attention.

Figure 4: Regions A,B,C,D should be labeled on both subplots (a) and (b). – We have added letters A-D on Figure 4a.

Also, it's not very clear how the amplitude plot in (a) was obtained. As I understand it, you extracted the outline of the glacier drainage network based on high reflection amplitudes observed in the data (blue lines in Figure 3). Then you went along this identified drainage network and calculated the RMS amplitude in a 2-m window centered around the drainage network (?) If this is the case, then why do we see only a thin yellow zone with high reflection amplitudes in (a)? Wasn't this entire drainage network region chosen because of high amplitudes in the data? - You are correct, the main drainage network was identified in the GPR data and picked based upon it being high amplitude and a specular and continuous reflection. The reasoning why we only have a dominant yellow zone could be due to two reasons: 1. The dominant yellow amplitude in Fig 4a represents the centre of the channel and the location where melt water is flowing within the conduit. Whereas the blue areas (lower amplitude) surrounding this main channel are areas where the GPR returns a continuous reflection (albeit smaller amplitude than the main channel but higher than the surrounding noise). This could be the result of water infiltrating the ice near the conduit's walls (i.e., water saturated ice). The width of the channel has been derived from the yellow section of Fig 4 and therefore, our conclusion that the channel is thin and wide remains 2. A result of the horizontal resolution of the 25 MHz GPR data.

You also mention that the high-amplitude (yellow) regions here correspond to water, but how do you know? Couldn't they correspond to air in the channel? – We are displaying RMS amplitude in Figure 4a however, the polarity of the reflected GPR wavelet was identical to the bed reflection. In the case of air being in the conduit we would expect the opposite polarity from the bed; this was check across the entire survey site to ensure that a polarity flip is not observed and therefore, we can rule out the reflection being caused by the presence of air.

Finally, the RMS amplitudes in the south are very low (near zero), suggesting that a conduit is not present. What is happening there? – We can speculate that further down-glacier the conduit is becoming somewhat diffusive and is no longer a single main branch. However,

**hard conclusions are difficult to draw. This observation (D) has been updated in the text to state it encounters a diffusive network of englacial conduits towards to the terminus.**

Line 130: Observation "D" is not clear for me from Figure 4. – We have added more information, while addressing the previous comment.

Lines 132-142: This paragraph attempts to use the GPR results in Figure 4a to assess the width and height of the identified channels, but for me the statements are far too conclusive given the resolution limitations and uncertainties in the data, and require some important assumptions. For example, in your assessment of the channel height, you appear to make use of the 1/4 wavelength vertical resolution criterion, which at 25 MHz and for water (velocity = 0.033 m/ns) is around 0.33 m. But this assumes that the channel is water-filled, which may not be the case. As the identified channels are extremely large, couldn't they be at least partly filled with air? In the case of an air-filled channel, the 1/4 wavelength value increases to 3 m meaning the channel height could be much greater. With regard to horizontal resolution, the GPR wavelength in ice will also have some effect. In perfectly migrated data, for example, the limit to horizontal resolution (if I remember correctly) is 1/2 wavelength, which for ice and 25 MHz is 0.84 m. But practically the value will be greater than this because of the limited migration aperture, lack of taking into account antenna radiation patterns, constant velocity assumption, etc. In short, I think some detailed discussion on resolution is needed in the manuscript, and statements should be written to reflect the substantial uncertainty as a result of limits to resolution. - We have updated the manuscript regarding the uncertainties in the horizontal direction based upon the horizontal resolution (wavelength in ice / 4 = 1.67 m). Furthermore, we have also discussed the assumption that the conduit is water-filled and therefore is at the limit of the vertical resolution. Should the conduit be air-filled the amplitude will be opposite to the bed reflection and this was not the case in our data. All wavelets identified as the drainage network exhibited the same polarity that resembles the ice-bed interface (albeit with different amplitudes) meaning that there is an increase in EM wave propagation velocity and thereby, ruling out an ice-air interface.

Lines 148 and 150: The word "indicating" tells me that you are sure, whereas it seems that there is some uncertainty in this interpretation (i.e., other things could explain higher amplitudes in the bed reflection, as mentioned above). I would replace with "which may indicate". – Updated manuscript as per suggestion.

Line 162, "A 3D migration effectively collapses...": The migration does indeed collapse the Fresnel zones and improve the resolution of the data, but I don't think it reduces it to the bin size (lateral resolution must still depend on wavelength, as noted above). We could not, for example, collect 25 MHz data with an extremely small bin size in all directions and have some limitless improvement in horizontal resolution. – Many thanks for pointing out this incorrect definition. This has been updated as per your suggestion to state the horizontal resolution shrinks to between ½ and ¼ wavelength post-migration.

Figure 5: How are you sure that none of the identified high-amplitude features are airfilled, which would also generate a strong bed reflection? Some explanation or justification is needed in the text. - We have added a section in the discussion relating to the fact that there

**were no air-filled cavities present as a result of the consistent reflected EM wave's phase across the basal interface.**

Line 173, "This is the first time that a glacier's drainage network is imaged in 3D with GPR data": Given the existing literature, I think that this is over-selling things a bit and should be modified. Harper et al. (2010) and Egli et al. (2021) used similarly dense 3D acquisitions to image elements of the drainage system, whereas Hansen et al. (2020) used 3D data (albeit at a greater line spacing) to characterize the drainage network in 3D. – We acknowledge this fact and have updated to state alongside Harper and Egli this is one of the first times that a glacier's network is imaged in 3D with GPR data.

Lines 183-184: Again, the rather extreme width-to-height ratio was derived under the assumption of a water-filled channel, which must be justified or stated as an assumption. – This is assumed however; we have evidence that it is water filled from borehole observations in 2018 (Church et al 2020) and the polarity of the reflected GPR wavelet. This part of the manuscript has been updated with this change.

Line 199: Replace "evidence" with "possible evidence" to reflect that it's not certain that it's subglacial water accumulation. – Added ' possible' to text.

Line 207, "The high amplitude reflections along the basal interface (Fig 5a) represent water accumulations...": Again, this sentence conveys an absolute certainty, whereas it seems that there is some uncertainty. – Have added some uncertainty into the wording by stating it is likely the result of water accumulations.

Line 233, "We found the dimensions of the conduit were 60 times wider than its thickness...": See previous comments. This is assuming a water-filled channel which, given the channel size, may not be the case (?) – See comment above

Line 237: Replaced "indicated" with "suggested". – Changed manuscript as requested.

Line 240 to end: For the reader, this paragraph suggests that this is the first application of high-resolution 3D GPR data to image glaciers, which is not the case. Please modify accordingly. For example: "
[revised manuscript text omitted]

---

## Author Response (AR1)

Dear Dr. MacGregor,

Thank-you for your positive feedback on the revised MS and your comments highlighting the additional minor changes. We have addressed your comments in the revised MS and with our comments below in red.

Best regards,
Gregory Church and all co-authors
* * *
27-45: Consider citing Putzig et al. (2018, https://www.sciencedirect.com/science/article/pii/S0019103517302543) here as an extraterrestrial example of 3-D radar imaging. → We have added this reference along with some relevant text to the Introduction.

50: "multiple 2D GPR" Missing word? → Added "multiple 2D GPR profiles"

53-55: Note that this Shreve hydropotential will only be accurate if a substantial vertical fracture network does not further complicate englacial transport, e.g., Gulley et al. (2017, https://www.cambridge.org/core/journals/journal-of-glaciology/article/effect-of-discrete-recharge-by-moulins-and-heterogeneity-in-flowpath-efficiency-at-glacier-beds-on-subglacial-hydrology/9734138C4BFC1AE1CF9128B241E60992). That is of course presumably the case for your field site given that you were able to survey from the ground safely. → Thanks for this comment.

109: m^2 → We are a little confused on this change as there is no "m" on this line. On line 110 there is a reference to a distance at 2 m, however this is the distance between line spacing and therefore, 2 m is correct.

210-215: All fair points. Here it might be worth considering Holschuh et al. (2020, https://pubs.geoscienceworld.org/gsa/geology/article-abstract/48/3/268/579962/Linking-postglacial-landscapes-to-glacier-dynamics) as an example of in-between 2-D and 3-D acquisition that permits finer resolution of cross-track bed topography. → Have added this reference along with an appropriate sentence.

231: this study is → Updated

237: Here I think the "-ice" is extraneous and "cold" should be replaced with "polar". → Updated

244: "a maximum of" can be removed without loss of meaning. → Updated

260-263: Break up second sentence…a bit hard to follow. → Updated

275: likely represent → Updated

320: "interconnectability"…an unusual term…how about "continuity"? → Updated

Other comments:

Furthermore, we have updated the movies as per your pre peer-review request:
1. Title has been updated on elevation slices to state 'Elevation slice'
2. We have overlaid a black line representing the glacier surface.
3. Finally, we think that the bedrock reflection is difficult to observe on the elevation slices due to the fact that the bedrock is varying so much and is not represented on a single elevation slice. Therefore, the reflection is present at different elevation slices and is difficult to track in such slices.